# NSF-mediated disassembly of on- and off-pathway SNARE complexes and inhibition by complexin

Ucheor B Choi[1,2,3,4,5], Minglei Zhao[6], K Ian White[1,2,3,4,5], Richard A Pfuetzner[1,2,3,4,5], Luis Esquivies[1,2,3,4,5], Qiangjun Zhou[1,2,3,4,5], Axel T Brunger[1,2,3,4,5]*

[1]Department of Molecular and Cellular Physiology, Stanford University, Stanford, United States; [2]Department of Neurology and Neurological Sciences, Stanford University, Stanford, United States; [3]Department of Structural Biology, Stanford University, Stanford, United States; [4]Department of Photon Science, Stanford University, Stanford, United States; [5]Howard Hughes Medical Institute, Stanford University, Stanford, United States; [6]Department of Biochemistry and Molecular Biology, University of Chicago, Chicago, United States

**Abstract** SNARE complex disassembly by the ATPase NSF is essential for neurotransmitter release and other membrane trafficking processes. We developed a single-molecule FRET assay to monitor repeated rounds of NSF-mediated disassembly and reassembly of individual SNARE complexes. For ternary neuronal SNARE complexes, disassembly proceeds in a single step within 100 msec. We observed short- (<0.32 s) and long-lived ($\geq$0.32 s) disassembled states. The long-lived states represent fully disassembled SNARE complex, while the short-lived states correspond to failed disassembly or immediate reassembly. Either high ionic strength or decreased αSNAP concentration reduces the disassembly rate while increasing the frequency of short-lived states. NSF is also capable of disassembling anti-parallel ternary SNARE complexes, implicating it in quality control. Finally, complexin-1 competes with αSNAP binding to the SNARE complex; addition of complexin-1 has an effect similar to that of decreasing the αSNAP concentration, possibly differentially regulating cis and trans SNARE complexes disassembly.
DOI: https://doi.org/10.7554/eLife.36497.001

*For correspondence:
brunger@stanford.edu

## Introduction

Soluble N-ethylmaleimide-sensitive factor attachment protein receptor (SNARE) proteins are essential for cellular processes including membrane trafficking, neurotransmitter release, and hormone secretion (*Rothman, 2014*; *Südhof, 2013*; *Wickner and Schekman, 2008*). While many of the molecules involved in these processes are known, the mechanistic details of many steps remain unclear. Here, we primarily focus on the neuronal SNAREs, synaptobrevin-2, SNAP-25A, and syntaxin-1A. SNAREs from opposing membranes assemble into a *trans* SNARE complex and provide the energy for membrane fusion upon zippering into a *cis* SNARE complex (*Sutton et al., 1998*; *Weber et al., 1998*). Subsequently, the *cis* SNARE complex is disassembled by the ATPase N-ethylmaleimide-sensitive factor (NSF) (*Block et al., 1988*; *Malhotra et al., 1988*) in combination with SNAP (soluble NSF attachment protein) adapter proteins (*Weidman et al., 1989*). While there are three SNAP isoforms—αSNAP, βSNAP, and γSNAP (*Clary et al., 1990*; *Nishiki et al., 2001*; *Söllner et al., 1993a*; *Whiteheart et al., 1992*; *Wilhelm et al., 2014*)—most in vitro studies have been carried out with αSNAP. Prior to ATP hydrolysis and together with SNAPs and the SNARE complex, NSF forms the so-called 20S complex; the structural details of the starting point for this process have been described previously (*Zhao et al., 2015*; *White et al., 2018*). NSF and the SNAPs also disassemble

other configurations of the SNARE complex, such as the binary complex consisting of the t-SNAREs syntaxin-1A and SNAP-25A (*Ma et al., 2013*). Combined with the essential synaptic proteins Munc13 and Munc18, these four proteins comprise an essential system that produces and maintains a pool of fusogenic *trans* SNARE complexes (*Lai et al., 2017*; *Ma et al., 2013*).

In addition to NSF and SNAPs, several proteins interact with SNAREs, including complexin (Cpx – we primarily focus on the complexin-1 isoform), synaptotagmin-1 (*Zhou et al., 2017*), Munc18 (*Misura et al., 2000*), and Munc13 (*Lai et al., 2017*). Among these proteins, Cpx has by far the highest affinity for the full-length ternary SNARE complex ($K_D \approx$ 10–100 nM, (*Choi et al., 2016*; *Li et al., 2007*; *Pabst et al., 2002*), higher than that of αSNAP for the SNARE complex ($K_D \approx$ 1.5 μM, (*Vivona et al., 2013*). Functionally, Cpx plays a critical role in neurotransmitter release by activating synchronous release upon the arrival of action potential in the synaptic terminal and by regulating spontaneous release (*Mohrmann et al., 2015*; *Trimbuch and Rosenmund, 2016*). Structurally, Cpx forms a bipartite complex with the ternary SNARE complex (*Chen et al., 2002*), and, together with the $Ca^{2+}$ sensor synaptotagmin, it forms a tripartite complex that represents a primed prefusion state of these synaptic proteins (*Zhou et al., 2017*). The high affinity of Cpx for the SNARE complex relative to αSNAP raises the question as to whether Cpx is involved in regulating the disassembly activity of NSF.

A variety of bulk assays have been developed to study NSF-mediated SNARE complex disassembly (*Cipriano et al., 2013*; *Lauer et al., 2006*; *Pabst et al., 2000*; *Söllner et al., 1993b*; *Winter et al., 2009*). While informative and relatively easy to carry out, these assays provide only limited information about the reaction mechanism of a multicomponent system such as the 20S complex. Single-molecule fluorescence resonance energy transfer (smFRET) experiments overcome this limitation by revealing complex dynamics hidden in bulk measurements, enabling identification of individual dynamics of components or domains of the proteins as they participate in the global reaction (*Roy et al., 2008*). Such smFRET experiments are thus a powerful tool to study the dynamics of NSF-mediated disassembly, as the reaction involves many moving parts undergoing large conformational changes over the course of the reaction. A previous single-molecule study monitored at most one disassembly event per individual SNARE complex (*Ryu et al., 2015*). We developed an new single-molecule approach using a linked SNARE complex where the three SNARE proteins are fused by long, flexible linkers (*Gao et al., 2012*; *Zorman et al., 2014*) to enable monitoring of multiple rounds of SNARE complex disassembly and reassembly.

Here, we present results that fall into two categories. First, with respect to the development of the smFRET disassembly assay with linked SNAREs, we validate the structure of the 20S particle with linked SNAREs by cryo-EM and demonstrate that measured kinetic parameters are consistent with those obtained previously. Factors known to influence the disassembly reaction, such as ionic strength and αSNAP concentration, are explored. Second, having validated the assay, we then use it to characterize the disassembly process under new conditions. Critically, we find that off-target, antiparallel SNARE complexes and binary (t-SNARE) complexes are disassembled at a faster rate than properly assembled ternary SNARE complexes. Finally, the addition of the high-affinity binding factor Cpx partially interferes with NSF-mediated disassembly.

## Results

### The linked SNARE complex does not interfere with 20S complex formation

We expressed synthetically-linked SNARE complex (L-SNARE) as a single polypeptide with all endogenous cysteines mutated to serines (*Gao et al., 2012*; *Zorman et al., 2014*) (*Figure 1A*) and introduced specific cysteine mutations for labeling purposes (Materials and methods). To ensure that the synthetic linkers Sp1 and Sp2 in the L-SNARE complex do not interfere with or alter the structure of the 20S complex, we determined a cryo-EM structure using the L-SNARE complex (referred to as L-20S complex, *Figure 1B–C*, *Table 1*). Two key differences distinguish this structure from the previous structure of the 20S complex (*Zhao et al., 2015*)—first, the use of full-length SNAP-25A instead of two SNAP-25A fragments, and second, the presence of the synthetic linkers of the L-SNARE complex.

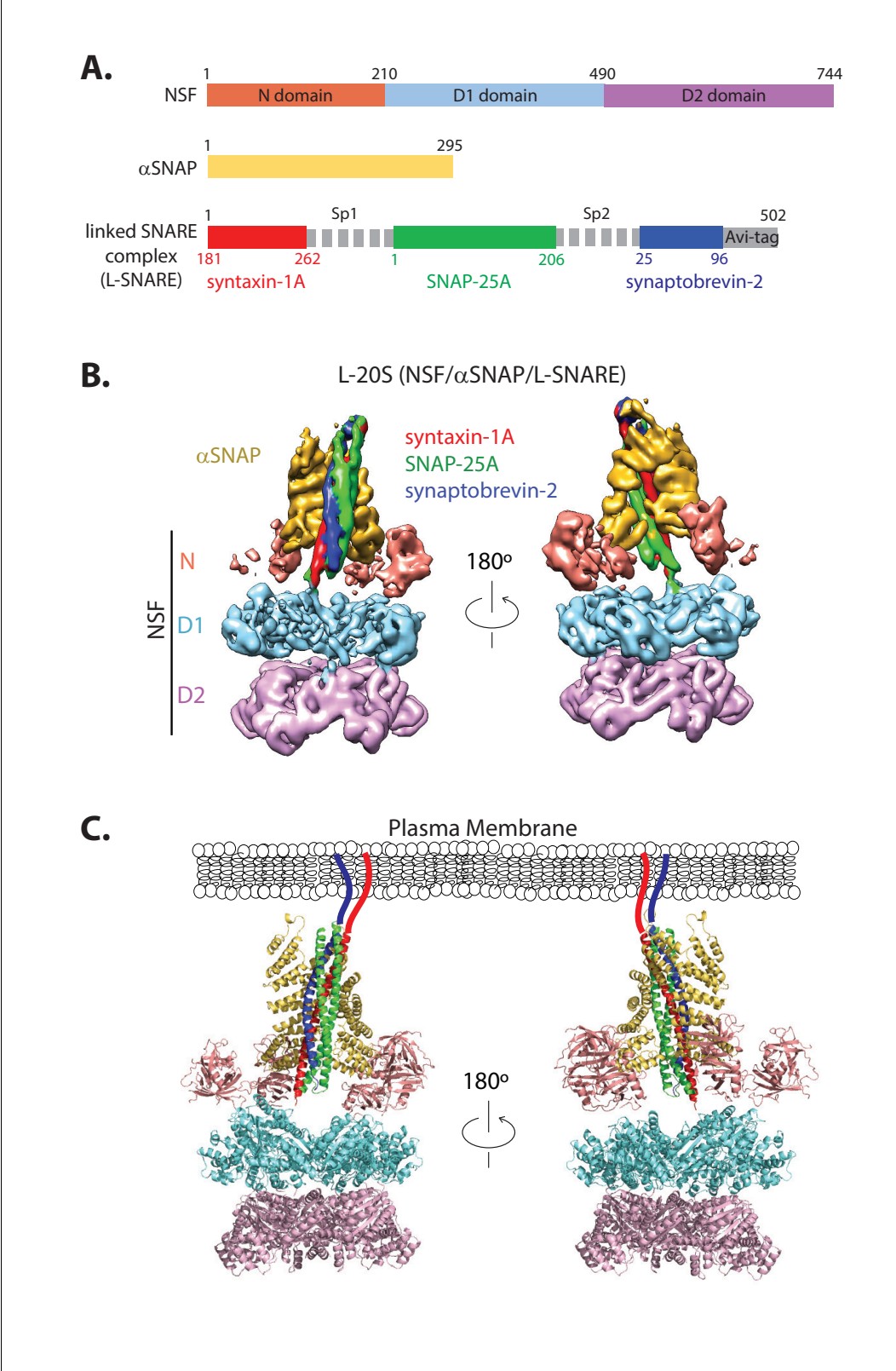

**Figure 1.** Structure of the L-20S (NSF/αSNAP/L-SNARE) complex. (A) Domain diagram of the L-20S complex consisting of NSF (orange, light blue, pink), αSNAP (yellow), and the L-SNARE complex composed of the cytoplasmic domain of syntaxin-1A (red, amino acid range 181–262), full-length SNAP-25A (green), and the cytoplasmic domain of synaptobrevin-2 (blue, amino acid range 25–96) fused by the synthetic linkers Sp1 and Sp2 and a biotinylation sequence (Avi-tag) at the C-terminus (Materials and methods). (B) Two orthogonal side views of the cryo-EM map of the L-20S complex

*Figure 1 continued on next page*

*Figure 1 continued*

filtered to a resolution of 7.0 Å and sharpened with a B-factor of −225 Å$^2$ (*Table 1*). The color code remains the same as in panel A. (C) Two orthogonal views of the three-dimensional model of the L-20S complex.

DOI: https://doi.org/10.7554/eLife.36497.002

The following figure supplements are available for figure 1:

**Figure supplement 1.** Single particle cryo-EM analysis.

DOI: https://doi.org/10.7554/eLife.36497.003

**Figure supplement 2.** Protein quantification of the 20S complex using a gel-based assay.

DOI: https://doi.org/10.7554/eLife.36497.004

The reconstructed density map of the L-20S complex has an estimated resolution of 7.0 Å without imposing any symmetry after refinement with RELION (*Scheres, 2012*) (*Figure 1—figure supplement 1*). The structure of the L-20S complex is similar to a recent high-resolution structure of the 20S complex prepared with full-length soluble SNAP-25A and full-length soluble syntaxin-1A but without synthetic linkers between the SNARE proteins (*White et al., 2018*). Both structures show two bound αSNAP molecules rather than four as seen in the structure of the 20S complex lacking the native linker between the two SNAP-25A SNARE motifs (*Zhao et al., 2015*). This reduction of the number of αSNAP molecules bound to the 20S complex was also confirmed by a gel-based assay, where the 20S complex was formed in the presence and absence of the native linker of SNAP-25A (amino acid range 86–119) (*Figure 1—figure supplement 2A–B*). Consistent with the cryo-EM structures, the αSNAP:NSF protein ratio decreased by a factor of two when the native linker of SNAP-25A was included (*Figure 1—figure supplement 2C*).

Because the chemical identity of each polypeptide chain of the SNARE complex was not distinguishable in the EM density map of the L-20S complex, we used a superposition with the high-resolution structure of the corresponding 20S complex (*White et al., 2018*) to assign each SNARE motif in the complex (*Figure 1B–C*). In the resulting model, the two αSNAP molecules primarily bind to a portion of the surface of the SNARE complex formed by syntaxin-1A and synaptobrevin-2 (*Figure 1B*), leaving the surface formed by SNAP-25A largely accessible. Taken together, these similarities suggest minimal structural perturbation due to the presence of the synthetic linkers.

## Characterization of the single-molecule disassembly assay

For the single-molecule experiments, we covalently attached fluorophores to the residues that had been mutated to cysteine and tethered these complexes to a passivated surface (*Figure 2A*). Unless stated otherwise, residue 249 of syntaxin-1A and residue 82 of synaptobrevin-2 were stochastically labeled with FRET donor and acceptor dyes (referred to as L-SNARE-CC). The proximity of the two labels leads to high fluorescence resonance energy transfer efficiency (FRET) for the fully assembled parallel SNARE complex (*Sutton et al., 1998*). The L-SNARE-CC molecules were biotinylated in vivo and purified using a previously published protocol which includes a 7.5 M urea wash to remove improperly assembled SNARE complexes (*Choi et al., 2016*; *Lai et al., 2017*; *Weninger et al., 2003*). The L-SNARE-CC molecule was then surface-tethered via a biotin-streptavidin linkage. Prior

**Table 1.** Data collection and processing statistics of the L-20S complex.

| Electron microscope | Titan Krios |
| --- | --- |
| Accelerating voltage (kV) | 300 |
| Defocus range (μm) | −1.5 — −3.0 |
| Electron dose (e⁻/Å$^2$) | 78 |
| Pixel size (Å) | 1.01 |
| Initial particles (No.) | 220,554 |
| Final particles (No.) | 45,194 |
| Map resolution (Å) | 7.0 |
| Map sharpening B-factor (Å$^2$) | −225 |

DOI: https://doi.org/10.7554/eLife.36497.005

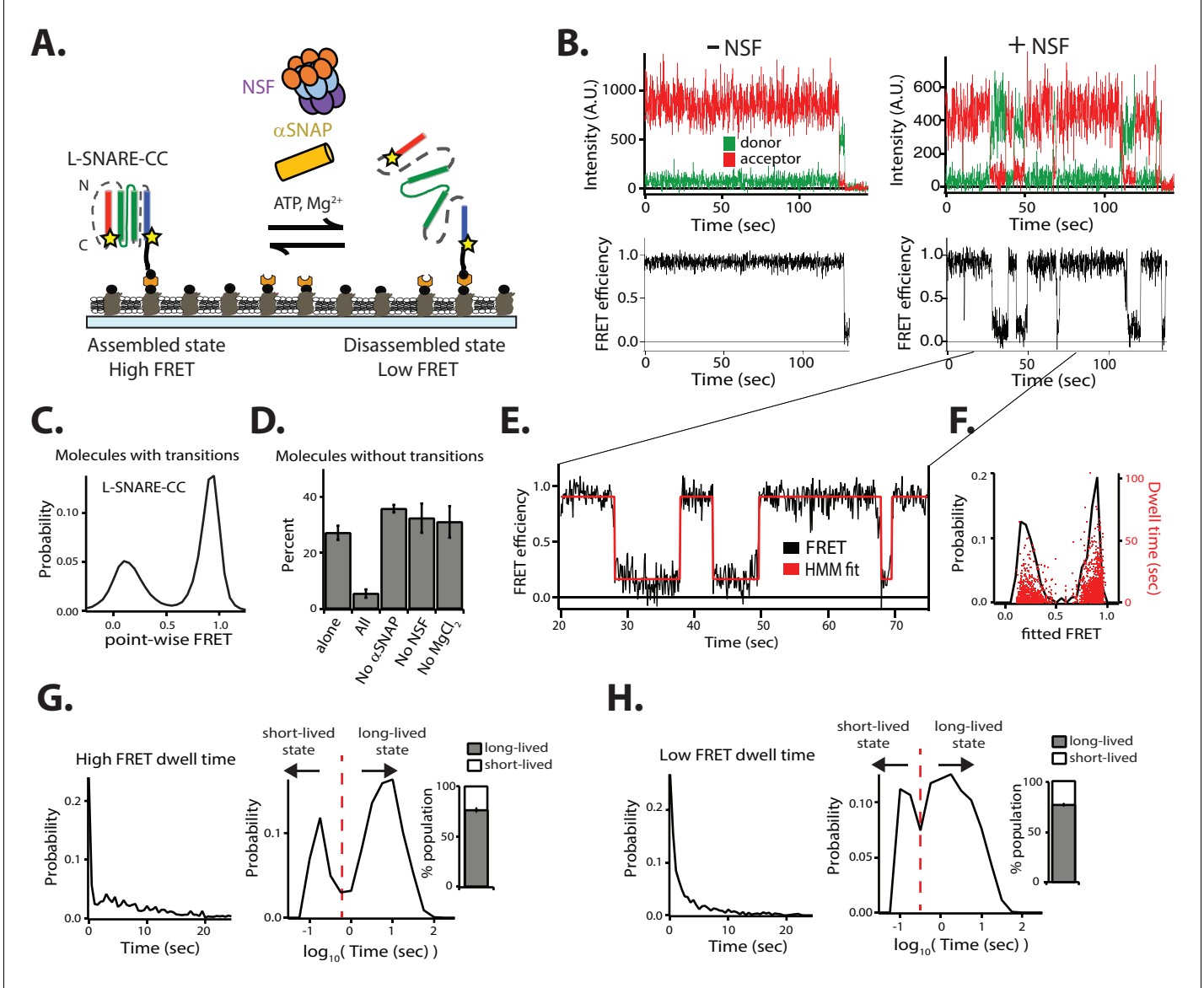

**Figure 2.** NSF-mediated disassembly and reassembly of single SNARE complexes. (**A**) Schematic of NSF-mediated disassembly of a single L-SNARE complex (colored as described in *Figure 1A*). Residues on syntaxin-1A and synaptobrevin-2 were stochastically labeled with fluorescent dyes at residues 249 and 82, respectively (indicated by stars) (referred to as L-SNARE-CC). The labeled L-SNARE-CC complex was surface tethered to a passivated surface and disassembly was subsequently initiated by adding the disassembly factors (Materials and methods). (**B**) Representative single-molecule fluorescence intensity time traces in the absence (top left panel) and presence (top right panel) of disassembly factors. The fluorescence intensities were converted to FRET efficiency time traces (bottom panels, Materials and methods). (**C**) Point-wise FRET efficiency histogram of all smFRET efficiency time traces of L-SNARE-CC molecules with transitions in the presence of disassembly factors. (**D**) Fraction of L-SNARE-CC molecules without transitions (mean values ± SD, Materials and methods, *Figure 2—source data 1*). (**E**) Expanded view of a smFRET efficiency time trace (black line) in the presence of disassembly factors in the selected time range (20–75 s), and a fit by HMM (red line, Materials and methods). (**F**) Probability distribution histogram of smFRET efficiencies (black line, left axis) and the corresponding dwell times (red dots, right axis) obtained from HMM. (**G–H**) smFRET efficiency dwell time histograms obtained from HMM on both linear and logarithmic timescales. Dashed red lines separate the short- and long-lived states (0.56 s for high FRET dwell times and 0.32 s for low FRET dwell times). Right subpanels show the populations (mean values ± SD) of the dwell times in the short- and long-lived states (Materials and methods, *Figure 2—source data 2*).

DOI: https://doi.org/10.7554/eLife.36497.006

The following source data and figure supplement are available for figure 2:

**Source data 1.** Data summary table for the results shown in *Figure 2D*.
DOI: https://doi.org/10.7554/eLife.36497.008
**Source data 2.** Data summary table for the results shown in *Figure 2G-H*.

*Figure 2 continued on next page*

*Figure 2 continued*

DOI: https://doi.org/10.7554/eLife.36497.009

**Figure supplement 1.** Representative smFRET efficiency time traces of L-SNARE-CC.

DOI: https://doi.org/10.7554/eLife.36497.007

to tethering, the surface was functionalized and passivated by biotinylated BSA and a surrounding phospholipid bilayer in order to prevent non-specific binding and to mimic a membrane environment (*Figure 2A*) (*Choi et al., 2012*).

We measured individual fluorescence intensity traces of the donor and acceptor dyes and calculated the smFRET efficiency for colocalized donor/acceptor pairs. Anti-correlated steps in the fluorescence intensity time traces indicate either transitions of the conformation of the L-SNARE-CC molecule or photobleaching of the acceptor dye (*Figure 2B*). Photobleaching events were discarded by inspection of the individual traces and omitted from the subsequent analysis of transitions in the smFRET efficiency time traces (Materials and methods, *Figure 2—figure supplement 1*). This is an advantage over a previous smFRET assay as performed with unlinked SNARE complexes (*Ryu et al., 2015*), which provides no means to readily distinguish photobleaching from disassembly. Moreover, our method does not require precise synchronization of the initiation of the disassembly process as we discard the first section of the observed smFRET efficiency time traces (Materials and methods, *Figure 2—figure supplement 1*).

In the presence of 1 μM NSF, along with 10 μM αSNAP, 1 mM ATP, and 1 mM MgCl$_2$ (referred to as 'disassembly factors'), multiple transitions typically occurred between high and low FRET efficiency states (*Figure 2B*, right panels). Multiple rounds of NSF-mediated disassembly and reassembly can be monitored for the same L-SNARE-CC molecule due to the presence of the synthetic linkers Sp1 and Sp2. The distribution of FRET efficiencies in the time traces (referred to as point-wise FRET) for the molecules that undergo transitions reveals two major populations, referred to as high FRET and low FRET, respectively (*Figure 2C*). These two populations are well-approximated by Gaussian distributions, so we inferred two distinct conformational states for the molecules. The high FRET efficiency state corresponds to the properly assembled configuration of the L-SNARE-CC molecule, while the low FRET efficiency state corresponds to the disassembled state following NSF action (*Figure 2C*). As control, if any or all the disassembly factors are omitted, constant high FRET efficiency is observed for all molecules (excluding photobleaching events) (*Figure 2B*, left panels, and *Figure 2D*).

To characterize the timescales and populations of the different L-SNARE-CC molecule states, we performed hidden Markov modeling (HMM) of the smFRET efficiency time traces of the molecules with transitions (*McKinney et al., 2006*) (*Figure 2E–H*). This particular method interprets the observed point-wise (i.e., time-binned) FRET efficiency traces in terms of as a hidden Markov process with the specified number of states. This method produces the most likely FRET values (based on a maximum likelihood approach) for these states for each individual molecule, and it also determines their interconversion rates and dwell times.

We assumed two states based on the observed point-wise FRET efficiency histogram (*Figure 2C*), resulting in two fitted FRET efficiencies from HMM for each of the individual molecules (*Figure 2E*, red line). The resulting probability distribution of the fitted FRET efficiencies (*Figure 2F*, black line) reveals two major populations (high FRET and low FRET), which are inferred from the point-wise FRET efficiency distribution (*Figure 2C*). The dwell times of the fitted FRET efficiencies are well-distributed in the two states (*Figure 2F*). To characterize the timescales of the different states, we plotted histograms of the high (*Figure 2G*) and low (*Figure 2H*) FRET efficiency dwell times. There are two distinct populations of dwell times in both histograms representing short- (<0.56 s for high FRET efficiency and <0.32 s for low FRET efficiency) and long-lived (≥0.56 s for high FRET efficiency and ≥0.32 s for low FRET efficiency) states. We thus calculated the populations for the four states and found that the long-lived states to be in the majority. Taken together, two distinct temporal states for both disassembly and reassembly of SNARE complexes are observed.

In order to calculate the disassembly rate for the L-SNARE-CC molecules that transition from the long-lived high FRET efficiency state, we fit a first-order exponential function to the high FRET efficiency dwell time histogram (*Figure 2G*, left panel) for times longer than 0.56 s, resulting in a rate of $1/(14.3 \pm 3.7)$ s$^{-1}$. This disassembly rate is comparable to that previously observed by smFRET

experiments using unlinked SNARE complexes reconstituted in liposomes ($1/14$ s$^{-1}$) (*Ryu et al., 2015*), and to that estimated from a bulk fluorescence dequenching assay (~$1/20$ s$^{-1}$) (*Cipriano et al., 2013*).

## High ionic strength decreases the disassembly activity

As previously reported, the activity of NSF/αSNAP is sensitive to the ionic strength of the solution (*Cipriano et al., 2013*; *Vivona et al., 2013*; *Zhao et al., 2015*). Indeed, in our smFRET assay, increasing the NaCl concentration led to a larger fraction of molecules that did not undergo transitions (*Figure 3A,C*). Moreover, the probability of low FRET efficiency decreased for the remaining molecules with transitions (*Figure 3B*), and the long-lived dwell times for high FRET efficiency increased (note the shift of the right maximum in *Figure 3D*), consistent with expectation. Finally, the populations of long-lived dwell times for low FRET decreased while the short-lived population increased (*Figure 3E*), (i.e., most occurrences of low FRET efficiency are single spikes in the time traces) (*Figure 3A*, right subpanel).

## Lowering the αSNAP concentration decreases the disassembly activity

NSF activity is highly dependent upon αSNAP concentration (*Clary et al., 1990*). For example, the disassembly rate in a bulk assay decreases as the αSNAP concentration is lowered (*Vivona et al., 2013*). In our smFRET assay, decreasing the αSNAP concentration had similar effects as increasing ionic strength (*Figure 3F–J*), as lowering the αSNAP concentration resulted in an increase in molecules without transitions (*Figure 3H*). Furthermore, the probability of low point-wise FRET efficiency decreased for the remaining molecules with transitions (*Figure 3G*), and the population of long-lived dwell times for low FRET efficiency decreased (*Figure 3J*). This decrease in NSF-mediated disassembly activity is expected considering the dissociation constant of 1.5 μM between αSNAP and the SNARE complex (*Vivona et al., 2013*); at 0.5 μM αSNAP, only 25% of the SNARE complex is expected to be bound to αSNAP, while at 20 μM, 93% is bound.

## Effect of αSNAP mutations on NSF-mediated disassembly

The effect of ionic strength on NSF-mediated disassembly of the SNARE complex suggests a critical role for charged residues in the formation of the 20S complex and/or the mechanism of disassembly. We tested this concept by mutating charged residues on αSNAP and assessed the effect on disassembly (*Figure 4A*). We examined a double mutant (K122E, K163E; referred to as KK mutant) designed to disrupt the interaction of αSNAP with the central part of the SNARE complex as well as a quadruple mutant (E39A, E40A, E43A, D80A; referred to as EEED mutant) designed to disrupt the interaction with the positive patch close to the C-terminus of the SNARE complex (*Figure 4A*) (*Zhao et al., 2015*). As expected, and consistent with previous results, the KK mutant of αSNAP abolished disassembly activity, while the EEED mutant had a milder effect (*Figure 4B–D*) (*Zhao et al., 2015*). Specifically, no transitions were observed for the KK mutant, while for the EEED mutant, the number molecules without transitions increased 2.5-fold compared to wildtype αSNAP (*Figure 4D*). Moreover, for the EEED mutant, the distribution of the long-lived dwell times for the high FRET efficiency state shifted to longer times (*Figure 4E*) while the population of short-lived dwell times for low FRET efficiency state increased and shifted towards shorter times (*Figure 4F*).

## Effect of the syntaxin Habc domain

Next, we investigated the effect of the syntaxin-1A Habc domain on the kinetics of NSF-mediated SNARE complex disassembly. We made a linked SNARE construct consisting of the full-length cytoplasmic domain of syntaxin-1A (amino acid range 1–263), full-length SNAP-25A, and synaptobrevin-2 (amino acid range 25–96) with the same labeling sites and linkers as used for the L-SNARE complex (termed L-SNARE$_{full}$-CC) (*Figure 5A*). The fraction of molecules without transitions and the point-wise FRET probability distributions are similar to those of the H$_{abc}$-free L-SNARE-CC construct (*Figure 5C–D*). Moreover, the short- and long-lived dwell time populations are similar (*Figure 5E–F*). Interestingly, however, the distributions of the long-lived dwell times for both high and low FRET efficiency shifted to longer times, indicating that the Habc domain reduces both disassembly and assembly rates of the SNARE complex (*Figure 5E–F*).

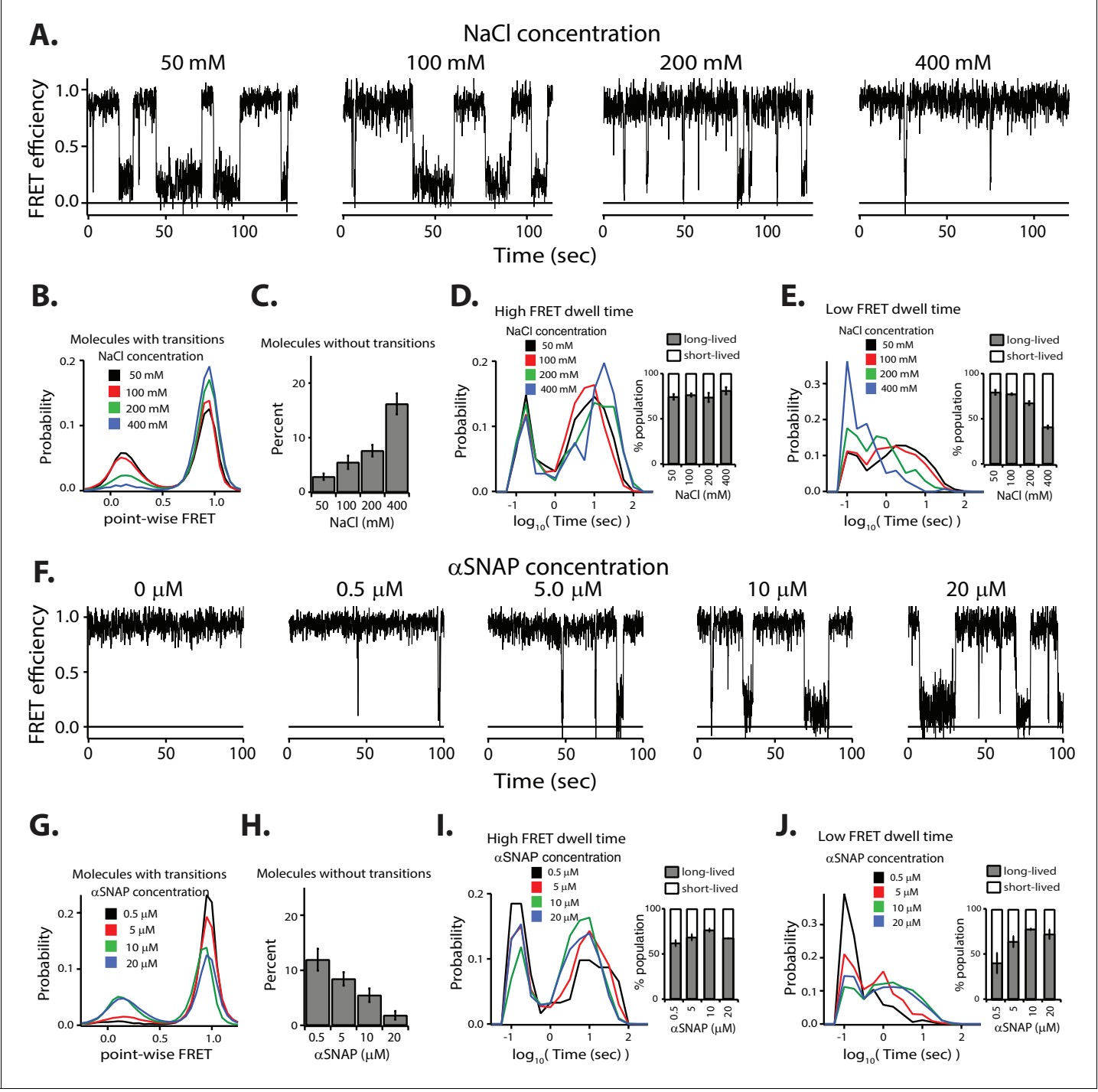

**Figure 3.** Increasing ionic strength or decreasing αSNAP concentration reduces NSF-mediated disassembly of L-SNARE-CC. (**A**) Representative smFRET efficiency time traces. (**B**) Corresponding point-wise FRET efficiency histograms using all observed time traces vs. NaCl concentrations. (**C**) Fraction of molecules without transitions (mean values ± SD, *Figure 3—source data 1*). (**D–E**) High and low smFRET efficiency dwell time histograms obtained from HMM. Right subpanels show the populations (mean values ± SD) of the dwell times in the short- and long-lived states (*Figure 3—source data 2*). (**F**) Representative smFRET efficiency time traces of NSF-mediated disassembly of L-SNARE-CC. (**G**) Corresponding point-wise FRET efficiency histograms using all smFRET efficiency time trace of molecules with transitions. (**H**) Fraction of molecules without transitions (mean values ± SD, *Figure 3—source data 3*). (**I–J**) smFRET efficiency dwell time histograms obtained from HMM. Right subpanels show the populations (mean values ± SD) of the dwell times in the short- and long-lived states (*Figure 3—source data 4*).
DOI: https://doi.org/10.7554/eLife.36497.010

The following source data is available for figure 3:

*Figure 3 continued on next page*

*Figure 3 continued*

**Source data 1.** Data summary table for the results shown in *Figure 3C*.
DOI: https://doi.org/10.7554/eLife.36497.011
**Source data 2.** Data summary table for the results shown in *Figure 3D-E*.
DOI: https://doi.org/10.7554/eLife.36497.012
**Source data 3.** Data summary table for the results shown in *Figure 3H*.
DOI: https://doi.org/10.7554/eLife.36497.013
**Source data 4.** Data summary table for the results shown in *Figure 3I-J*.
DOI: https://doi.org/10.7554/eLife.36497.014

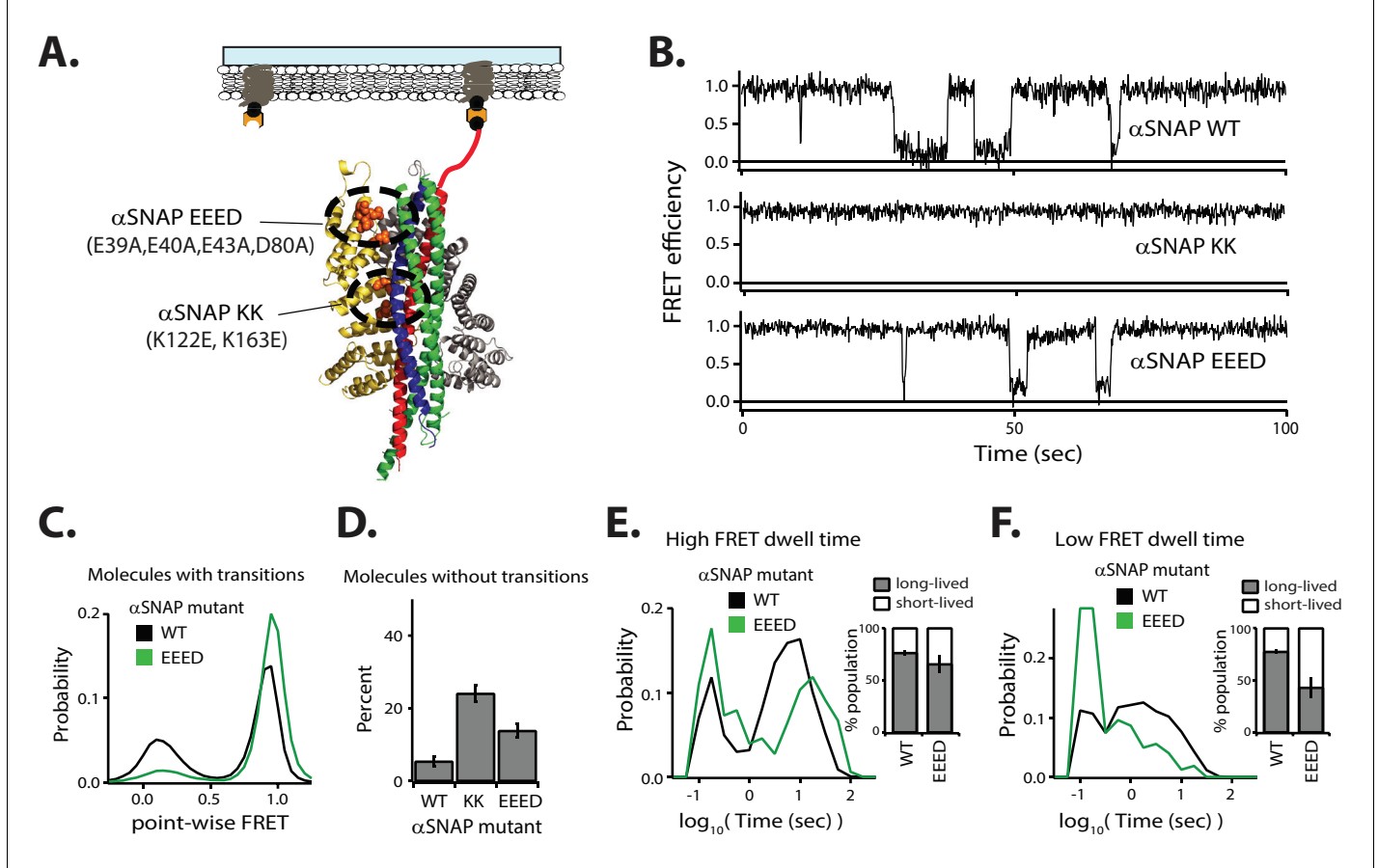

**Figure 4.** αSNAP mutants reduce NSF-mediated disassembly of L-SNARE-CC. (**A**) Structural schematic showing the locations of αSNAP mutations. (**B**) Representative smFRET efficiency time traces of L-SNARE-CC disassembly in the presence 10 μM wild type and mutant αSNAP. (**C**) Corresponding point-wise smFRET efficiency histograms using all smFRET efficiency time trace of molecules with transitions. (**D**) Fraction of molecules without transition (mean values ± SD, *Figure 4—source data 1*). (**E–F**) smFRET efficiency dwell time histograms obtained from HMM. Right subpanels show the populations (mean values ± SD) of the dwell times in the short- and long-lived states (*Figure 4—source data 2*).
DOI: https://doi.org/10.7554/eLife.36497.015

The following source data is available for figure 4:

**Source data 1.** Data summary table for the results shown in *Figure 4D*.
DOI: https://doi.org/10.7554/eLife.36497.016
**Source data 2.** Data summary table for the results shown in *Figure 4E-F*.
DOI: https://doi.org/10.7554/eLife.36497.017

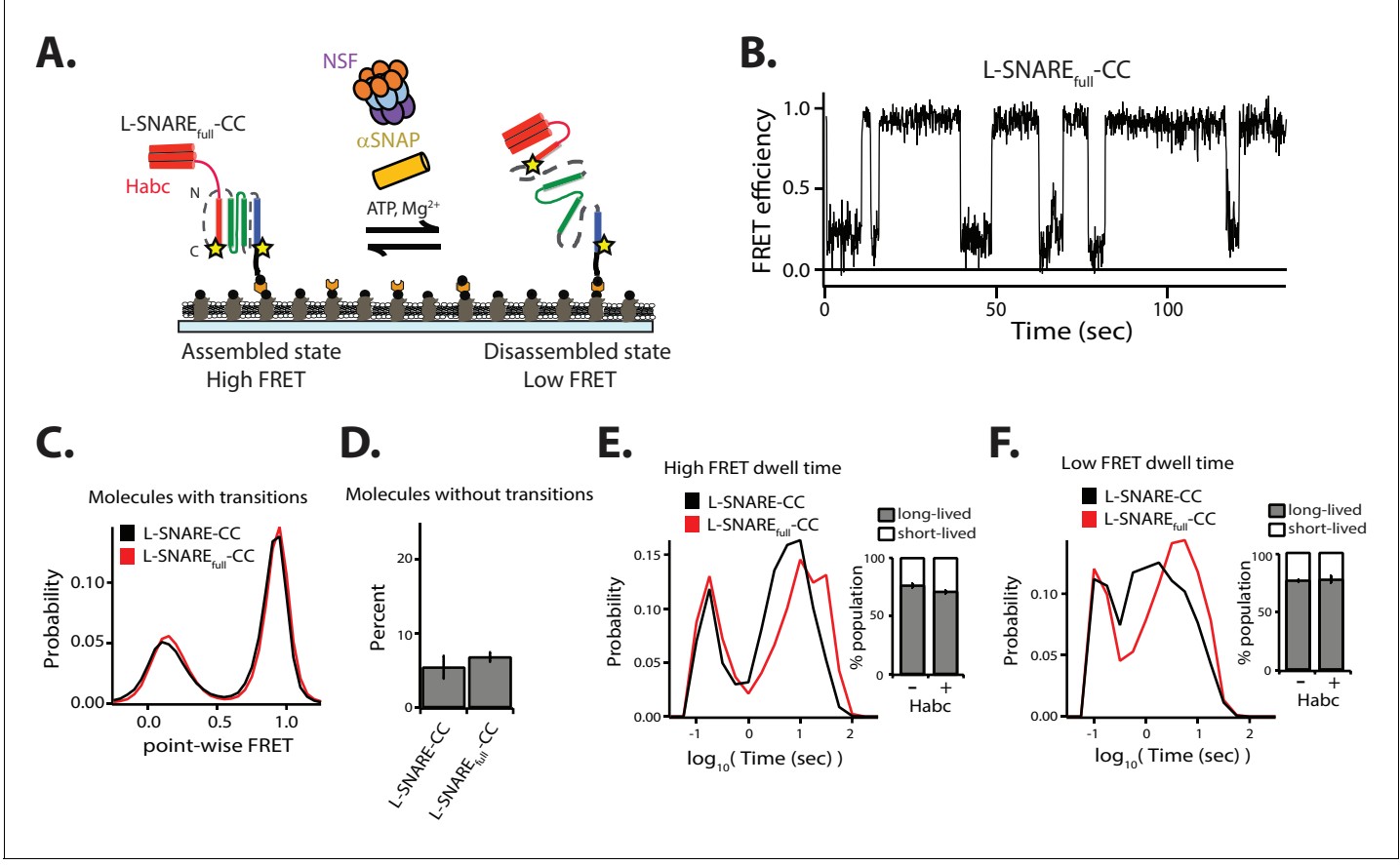

**Figure 5.** Effect of the N-terminal Habc domain of syntaxin-1A on disassembly. (**A**) Schematic of NSF-mediated disassembly of a single L-SNARE$_{full}$-CC complex labeled with fluorescent dyes (indicated by stars) at residue 249 of syntaxin-1A and residue 82 of synaptobvrevin-2. (**B**) Representative smFRET efficiency time traces for NSF-mediated L-SNARE$_{full}$-CC disassembly. (**C**) Corresponding point-wise smFRET efficiency histograms using all smFRET efficiency time trace of molecules with transitions. (**D**) Fraction of molecules without transitions (mean values ± SD, ***Figure 5—source data 1***). (**E–F**) smFRET efficiency dwell time histograms obtained from HMM. Right subpanels show the populations (mean values ± SD) of the dwell times in the short- and long-lived states (***Figure 5—source data 2***).

DOI: https://doi.org/10.7554/eLife.36497.018

The following source data is available for figure 5:

**Source data 1.** Data summary table for the results shown in ***Figure 5D***.
DOI: https://doi.org/10.7554/eLife.36497.019
**Source data 2.** Data summary table for the results shown in ***Figure 5E-F***.
DOI: https://doi.org/10.7554/eLife.36497.020

## Complexin interferes with NSF-mediated disassembly

We next investigated the effect of Cpx on the kinetics of NSF-mediated disassembly of single L-SNARE-CC molecules (***Figure 6A–F***). Overall, the effect of Cpx was similar to decreasing the αSNAP concentration or increasing the ionic strength of the disassembly buffer. Specifically, increasing the Cpx concentration resulted in an increase in molecules without transitions (***Figure 6D***). The probability of low point-wise FRET efficiency decreased for the remaining molecules with transitions (***Figure 6C***), and the population of long-lived dwell times for low FRET efficiency decreased (***Figure 6F***).

As control, we used the 4M mutant (R48A, R59A, K69A, Y70A) of Cpx, which disrupts the Cpx interaction with the SNARE complex (***Tang et al., 2006***). At 10 μM concentration, this mutant has intermediate effects on the point-wise FRET distribution and the number of molecules without transitions, while the dwell time distributions are similar to those without Cpx (***Figure 6B–F***).

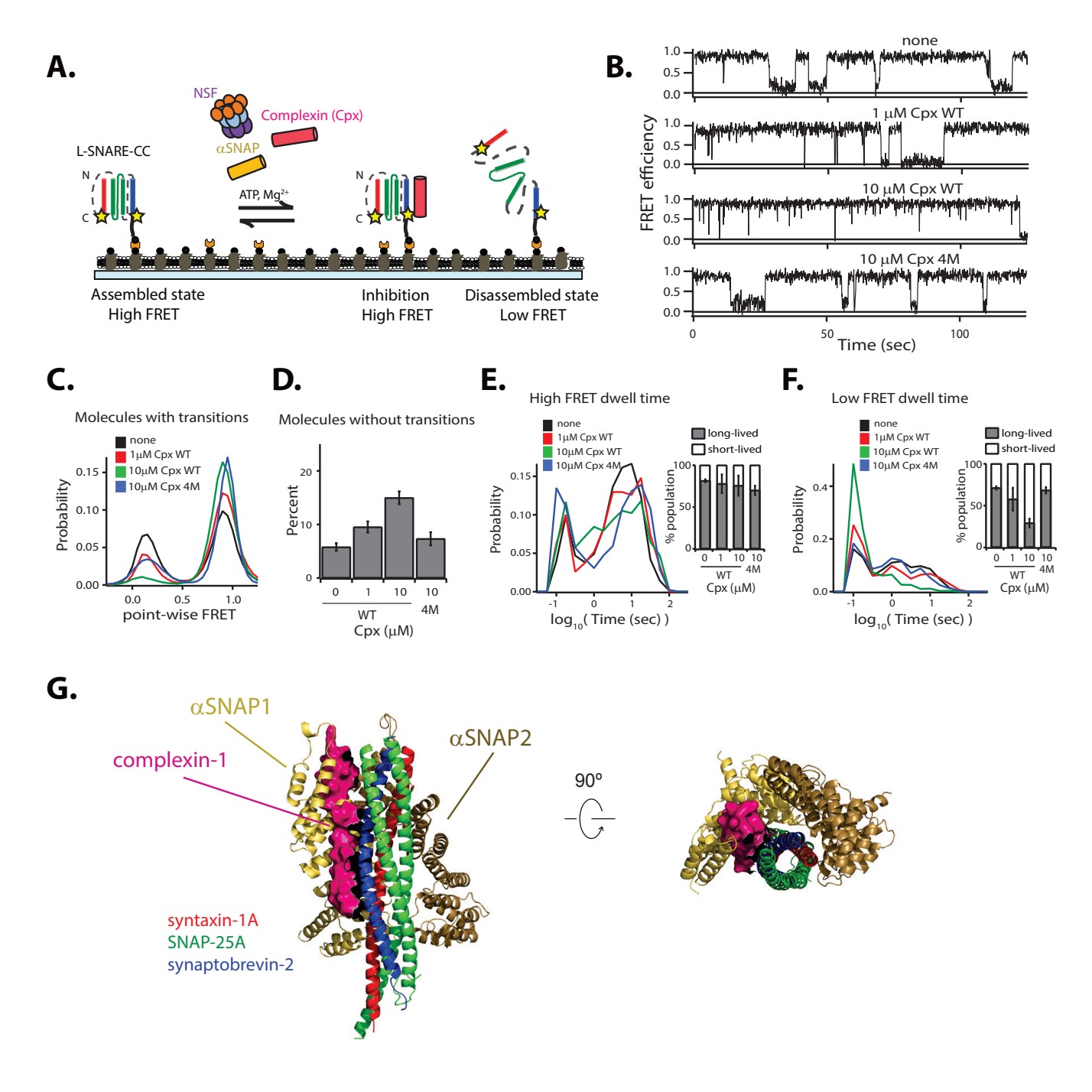

**Figure 6.** Cpx interferes with NSF-mediated disassembly of the SNARE complex. (**A**) Schematic of NSF-mediated disassembly of a single L-SNARE-CC complex labeled with fluorescent dyes (indicated by stars) at residue 249 of syntaxin-1A and residue 82 of synaptobvrevin-2 in the presence of Cpx. (**B**) Representative smFRET efficiency time traces corresponding to SNARE-CC disassembly in the absence or presence of 1 μM or 10 μM wildtype Cpx or 10 μM 4M mutant of Cpx. Concentrations of 1 μM and 10 μM Cpx correspond to a 10:1 and 1:1 molar ratio of αSNAP:Cpx, respectively. (**C**) Corresponding point-wise smFRET efficiency histograms using all smFRET efficiency time traces of molecules with transitions. (**D**) Fraction of molecules without transitions (mean values ± SD, *Figure 6—source data 1*). (**E–F**) smFRET efficiency dwell time histograms obtained from HMM . Right subpanels show the populations (mean values ± SD) of the dwell times in the short- and long-lived states (*Figure 6—source data 2*). (**G**) Superposition of the structure of the αSNAP/L-SNARE subcomplex (from the L-20S EM structure) with the crystal structure of the Cpx/SNARE complex (PDB ID: 1KIL). Cpx is shown as a surface and colored pink. The two αSNAPs are colored yellow and brown.

*Figure 6 continued on next page*

*Figure 6 continued*

DOI: https://doi.org/10.7554/eLife.36497.021

The following source data and figure supplement are available for figure 6:

**Source data 1.** Data summary table for the results shown in *Figure 6D*.

DOI: https://doi.org/10.7554/eLife.36497.023

**Source data 2.** Data summary table for the results shown in *Figure 6E-F*.

DOI: https://doi.org/10.7554/eLife.36497.024

**Figure supplement 1.** Competition of αSNAP and Cpx binding to the L-SNARE complex.

DOI: https://doi.org/10.7554/eLife.36497.022

To gain insight into how Cpx interferes with NSF-mediated SNARE complex disassembly, we performed a competition binding experiment (*Figure 6—figure supplement 1*). Cpx competes with αSNAP binding to the SNARE complex, with the most significant effect observed at Cpx:αSNAP molar ratios $\geq 1$. In contrast, the 4M mutant of Cpx has only a mild effect on αSNAP binding, comparable to the effect of wild type Cpx at a hundred-fold lower concentration (0.1 μM).

## NSF disassembles the binary t-SNARE complex

In addition to the ternary SNARE complex, NSF also disassembles the binary t-SNARE complex composed of syntaxin-1A and SNAP-25A (*Lai et al., 2017*; *Ma et al., 2013*). In order to understand the dynamics underlying this observation, we investigated the conformations and the kinetics of single binary t-SNARE complexes during NSF-mediated disassembly and reassembly. For our smFRET assay, we designed a construct by linking syntaxin-1A (amino acid range 181–262) and full-length SNAP-25A (*Figure 7A*). Two constructs were made with FRET labeling pairs at residue 249 of syntaxin-1A and either residue 76 or residue 197 of SNAP-25A; these constructs are referred to as either L-SNARE$_{binary}$-CC1 or L-SNARE$_{binary}$-CC2 molecules, respectively. Upon assembly into parallel helical bundles, both molecules are expected to produce high FRET efficiencies.

In contrast to the linked ternary SNARE complex L-SNARE-CC, we observed intermediate FRET efficiency in the point-wise probability distributions for both L-SNARE$_{binary}$-CC1 and L-SNARE$_{binary}$-CC2 (*Figure 7B–D*). We therefore fit a sum of three Gaussian distributions to the point-wise FRET efficiency distributions (*Figure 7C–D*), resulting in a reasonable fit supporting the notion of three distinct states (with low, mid, and high FRET efficiency) for the binary SNARE complex. The fraction of molecules without transitions for both L-SNARE$_{binary}$-CC1 and L-SNARE$_{binary}$-CC2 was similar (*Figure 7E*). We then fit the data using three-state HMM (*Figure 7F-I*). Compared to ternary L-SNARE-CC, the occurrence of long-lived dwell times decreased (red dots in *Figures 2F* and *7F–G*). The populations of the long-lived dwell times for both mid and low FRET efficiency decreased slightly (*Figure 7H–I*). As a control, the FRET efficiency time traces for both L-SNARE$_{binary}$-CC1 and L-SNARE$_{binary}$-CC2 did not show any conformational transitions (except for photobleaching events) in the absence of the disassembly factors (*Figure 7—figure supplement 1*) in agreement with previous observations. Furthermore, the distribution of point-wise FRET efficiencies is indicative of at least three conformations of the binary t-SNARE complex (fully assembled, or with either of the two SNAP-25A domains dislodged) (*Weninger et al., 2008*).

## Disassembly kinetics are largely independent of label site combinations

Having characterized the disassembly kinetics of the ternary SNARE complex, we next investigated if the results depend on the particular choice of labeling sites. We designed L-SNARE constructs with stochastic FRET labeling pairs at residue 249 of syntaxin-1A and residue 76 of the SNAP-25A SN1 domain (referred to as L-SNARE$_{ternary}$-CC1), and residue 249 of syntaxin-1A and residue 197 of the SNAP-25A SN2 domain (referred to as L-SNARE$_{ternary}$-CC2). Additionally, we designed a construct with stochastic FRET labeling pairs at residue 193 of sytnaxin-1A and residue 28 of synaptobrevin-2 (referred to as L-SNARE-NN) (*Figure 8A*). These labeling combinations are expected to produce high FRET efficiency in the properly assembled ternary SNARE complex.

As observed in the case of L-SNARE-CC in the presence of the disassembly factors, L-SNARE-NN, L-SNARE$_{ternary}$-CC1, and L-SNARE$_{ternary}$-CC2 all showed multiple transitions between high and low FRET efficiency. Differences in the point-wise FRET distributions can be explained by variations

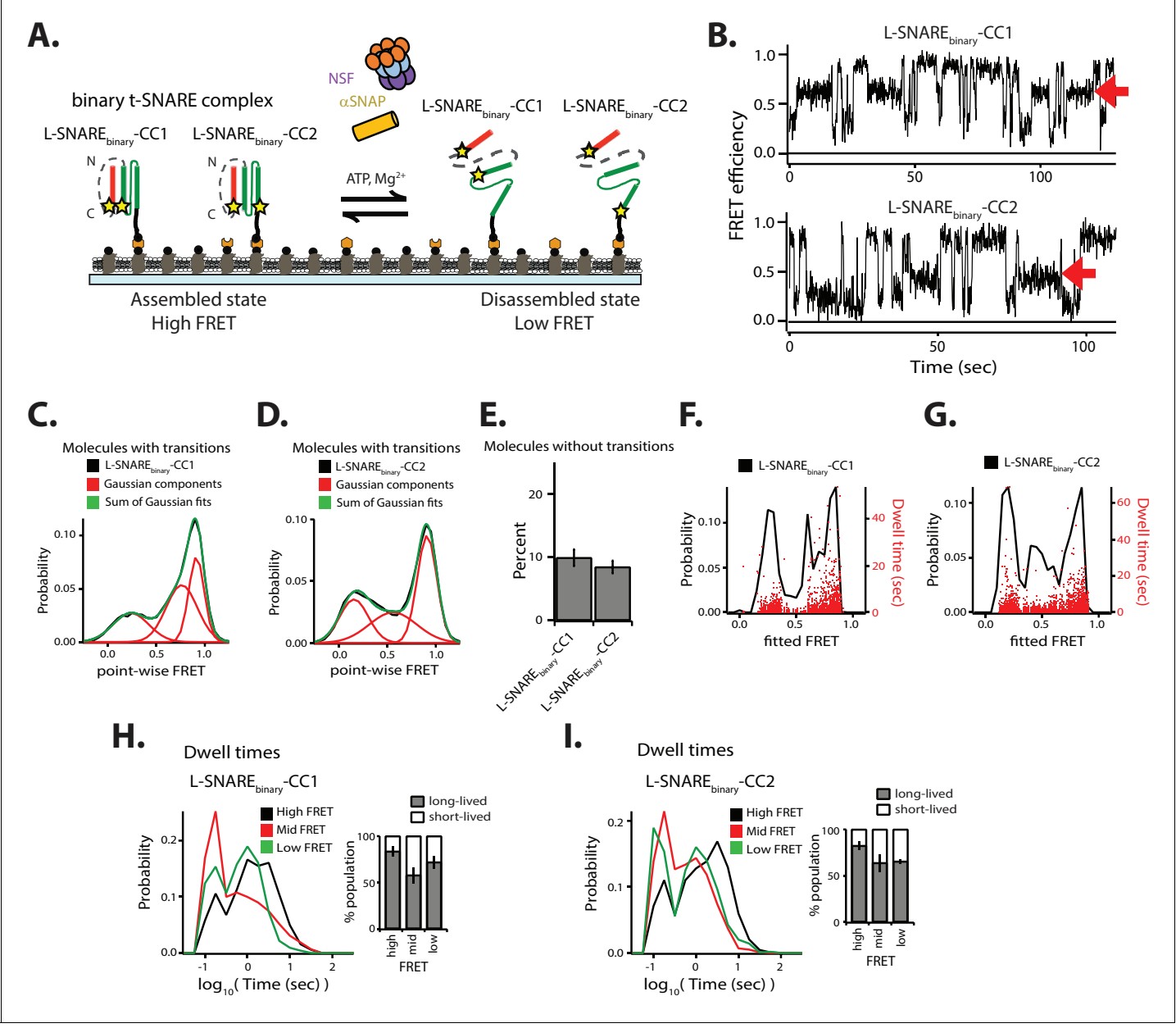

**Figure 7.** NSF-mediated disassembly of the binary t-SNARE complex. (A) Schematic of NSF-mediated disassembly of a single linked binary t-SNARE complex composed of syntaxin-1A and SNAP-25A. L-SNARE$_{binary}$-CC1 and L-SNARE$_{binary}$-CC2 were labeled (indicated by stars) at residue 249 of syntaxin-1A and at either residue 76 or 197 of SNAP-25A, respectively. (B) Representative smFRET efficiency time traces of L-SNARE$_{binary}$-CC1 (left panel) and L-SNARE$_{binary}$-CC2 (right panel). Red arrows indicate intermediate FRET efficiency states. (C–D) Corresponding point-wise smFRET efficiency histograms using all smFRET efficiency time traces of molecules with transitions. A sum of three Gaussian functions was fit to the histogram representing the distribution of the individual states (black: observed point-wise FRET; red: Gaussian components; green: sum of Gaussian fits). (E) Fraction of molecules without transitions (mean values ± SD, *Figure 7—source data 1*). (F–G) Probability distribution histogram of smFRET efficiencies (black line, left axis) and the corresponding dwell times (red dots, right axis) obtained from HMM. (H–I) smFRET efficiency dwell time histograms obtained from HMM. Right subpanels show the populations (mean values ± SD) of the dwell times in the short- and long-lived states (*Figure 7—source data 2*).

DOI: https://doi.org/10.7554/eLife.36497.025

The following source data and figure supplement are available for figure 7:

**Source data 1.** Data summary table for the results shown in *Figure 7E*.
DOI: https://doi.org/10.7554/eLife.36497.027
**Source data 2.** Data summary table for the results shown in *Figure 7H–I*.

*Figure 7 continued on next page*

*Figure 7 continued*

DOI: https://doi.org/10.7554/eLife.36497.028

**Figure supplement 1.** Configurations of isolated linked binary t-SNARE complexes.

DOI: https://doi.org/10.7554/eLife.36497.026

in distances for these labeling pairs in both the assembled and disassembled SNARE complexes (*Figure 8B–C*). There are also differences in the percentage of molecules without transitions (*Figure 8D*); these are likely due to variation in the efficiency with which 20S complex forms (i.e., some of the dye combinations may differentially affect the assembly of the 20S complex). Nevertheless, the dwell time distributions and the short- and long-lived populations are similar (*Figure 8E–F*). Thus, the kinetics of NSF-mediated SNARE complex disassembly are largely independent of the choice of FRET labeling pairs, enabling comparison of the dissembly of SNARE complexes labeled at different positions.

## NSF-mediated disassembly of improperly assembled SNARE complexes

When the SNARE proteins are simply mixed in the absence of Munc18 and Munc13, they are sometimes kinetically trapped in nonproductive (e.g., anti-parallel) configurations (*Choi et al., 2016*; *Lai et al., 2017*; *Sakon and Weninger, 2010*; *Weninger et al., 2003*). To observe these improperly assembled states in our system, we designed FRET pairs to report high FRET efficiency when configured in an anti-parallel or otherwise off-target state. The C-terminal labeling site at residue 249 on syntaxin-1A was kept the same as in previous experiments with L-SNARE-CC, but the labeling site on synaptobrevin-2 was moved to residue 28; this molecule (referred to as L-SNARE-CN) yields low FRET efficiency when the SNARE complex is properly assembled or disassembled, and it yields high FRET efficiency for anti-parallel configurations (*Figure 9A*). Consistent with previous observations (*Choi et al., 2016*; *Weninger et al., 2003*), molecules purifed without a urea wash were observed in both stable parallel and anti-parallel configurations (*Figure 9—figure supplement 1A,B*) in the absence of the urea wash. When the purification protocol included a urea wash step prior to surface tethering, L-SNARE-CN molecules were mostly found in the proper parallel configuration (*Figure 9— figure supplement 1A*).

In the presence of disassembly factors, the smFRET efficiency time traces of the anti-parallel-reporting L-SNARE-CN molecules showed brief transitions to high FRET efficiency corresponding to improperly assembled anti-parallel configurations (*Figure 9B–C*, *Figure 9—figure supplement 1C*). These improperly assembled SNARE-CN molecules were then rapidly disassembled by NSF as indicated by the subsequent disappearance of high FRET efficiency (*Figure 9B*), while they remained in the high-FRET efficiency configuration in the absence of disassembly factors (*Figure 9—figure supplement 1B*). Nearly 50% of the dwell times for high FRET efficiency states were short-lived (< 0.56 s, *Figure 9F*), indicating a rapid NSF-mediated disassembly of anti-parallel configurations. Taken together, our data show that NSF is capable of disassembling improperly assembled anti-parallel SNARE complexes on a timescale faster than that of properly assembled ternary SNARE complexes.

## Discussion

### Monitoring cycles of NSF-mediated disassembly and reassembly at the single-molecule level

Due to the complexities of this multi-component system, ensemble assays for exploring the molecular mechanism of NSF-mediated SNARE complex disassembly may provide only limited information. To overcome these limitations, we monitored the behavior of surface-tethered single SNARE complexes using FRET pairs that were covalently attached to different positions within the SNARE complex. Additionally, we studied a linked form of the SNARE complex, allowing us to observe multiple rounds of NSF-mediated disassembly and reassembly of the same molecule. This approach also distinguishes between true molecular events and events that are due to photophysics, such as photobleaching. As expected, the disassembly rates of the tethered and linked SNARE complexes (L-SNARE-CC molecules) obtained from our smFRET assay are similar to those measured in solution by other assays.

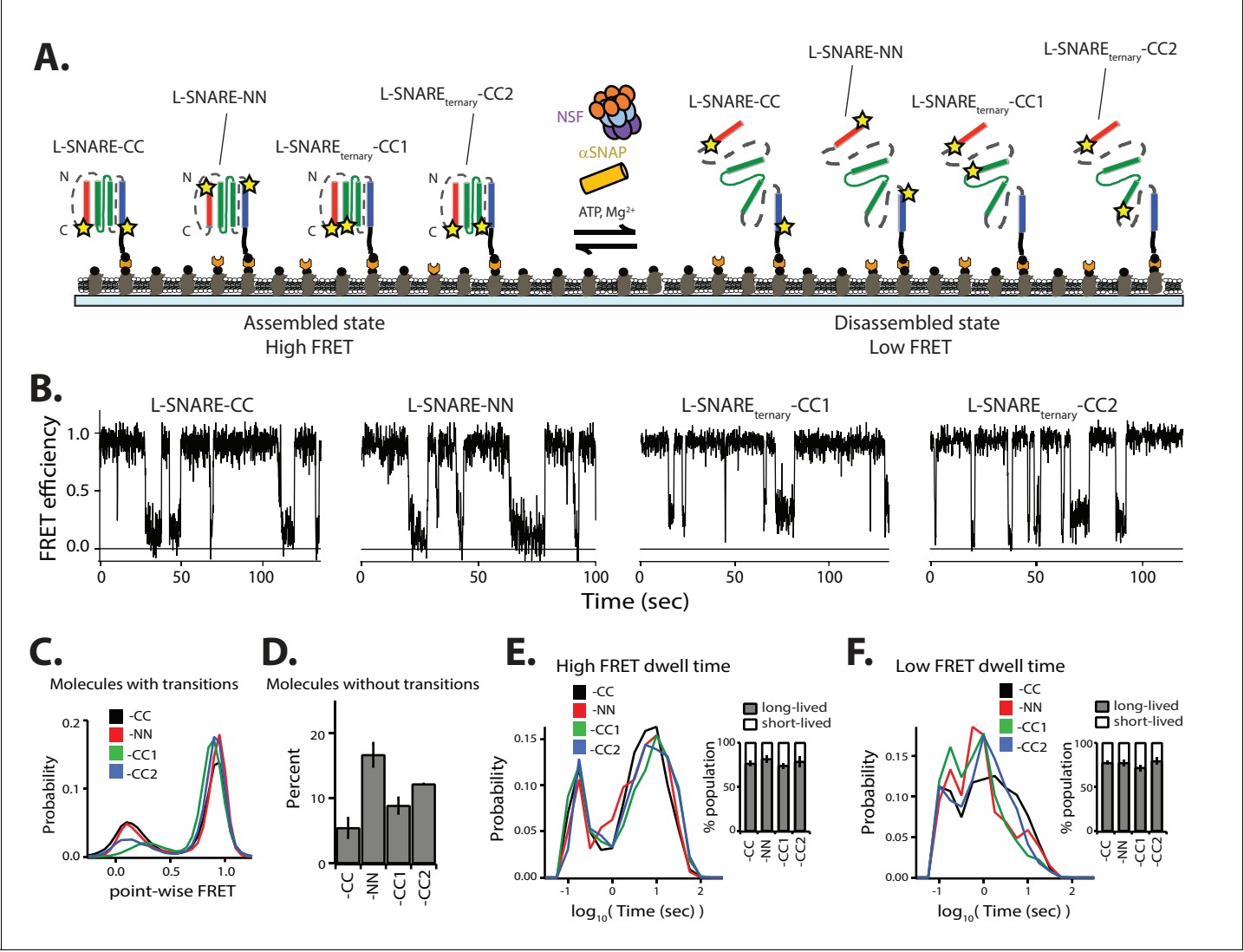

**Figure 8.** Different labeling combinations produce similar kinetics. (**A**) Schematic of NSF-mediated disassembly of a single L-SNARE-CC complex (labels at residue 249 of syntaxin-1A and residue 82 of synaptobrevin-2), L-SNARE-NN (labels at residue 193 of syntaxin-1A and residue 28 of synaptobrevin-2), L-SNARE$_{ternary}$-CC1 (labels at residue 249 of syntaxin-1A and residue 76 of SNAP-25A), and L-SNARE$_{ternary}$-CC2 (labels at residue 249 of syntaxin-1A and residue 197 of SNAP-25A). Stars indicate fluorescent dyes. (**B**) Representative smFRET efficiency time traces of NSF-mediated disassembly of L-SNARE-CC, L-SNARE-NN, L-SNARE$_{ternary}$-CC1, and L-SNARE$_{ternary}$-CC2. (**C**) Corresponding point-wise FRET efficiency histograms using all observed traces. (**D**) Fraction of molecules without transitions (mean values ± SD, *Figure 8—source data 1*). (**E–F**) smFRET efficiency dwell time histograms obtained from the HMM. Right subpanels show the populations (mean values ± SD) of the dwell times in the short- and long-lived states (*Figure 8—source data 2*).

DOI: https://doi.org/10.7554/eLife.36497.029

The following source data is available for figure 8:

**Source data 1.** Data summary table for the results shown in *Figure 8D*.
DOI: https://doi.org/10.7554/eLife.36497.030
**Source data 2.** Data summary table for the results shown in *Figure 8E–F*.
DOI: https://doi.org/10.7554/eLife.36497.031

For the first time, our smFRET assay reveals two temporal states for both SNARE complex disassembly and reassembly (*Figure 2G–H*), with the large majority of the SNARE complexes found in the long-lived states. With respect to disassembly, these long-lived states represent complete NSF-mediated disassembly, while the short-lived states are failed disassembly or disassembly followed by immediate reassembly. As for assembly, the short-lived state may correspond to a partially

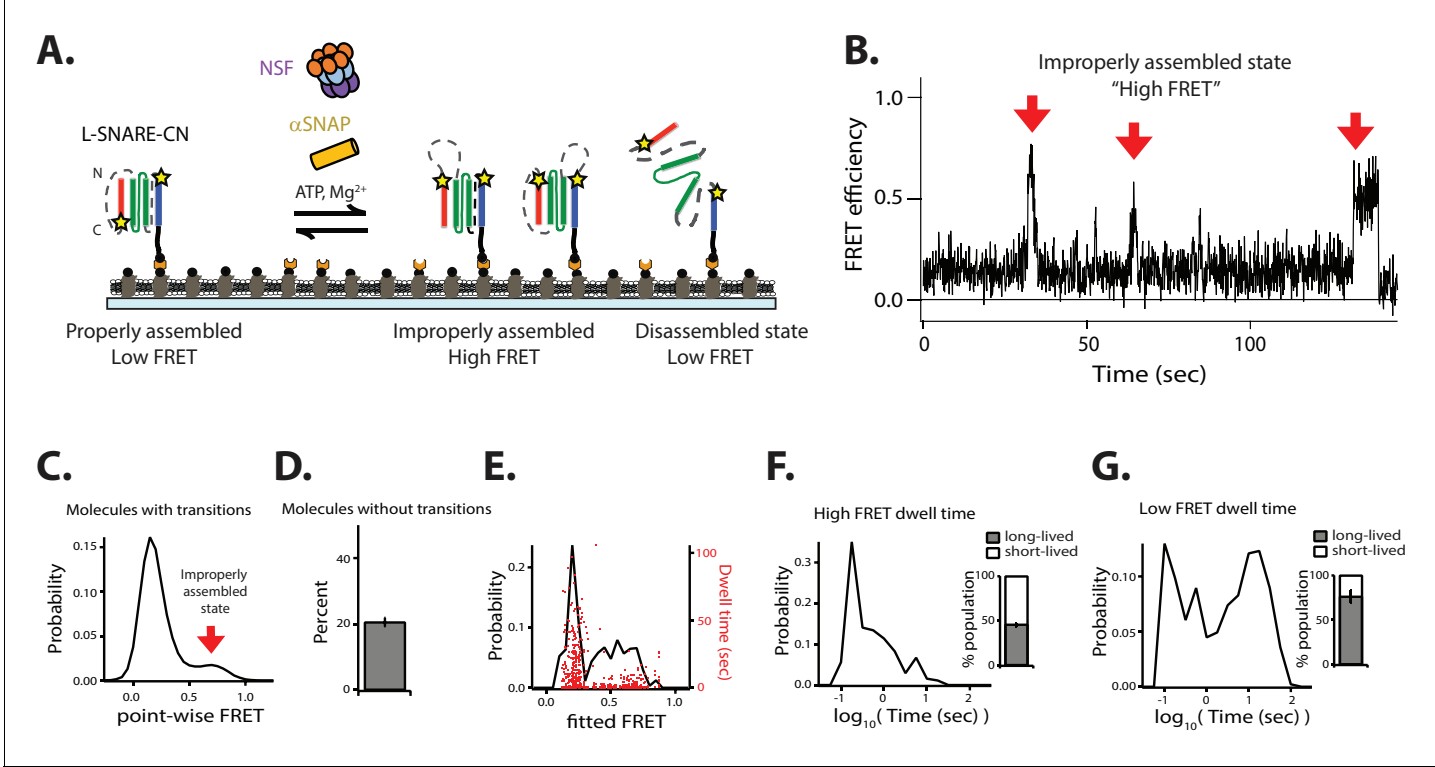

**Figure 9.** NSF disassembles anti-parallel L-SNARE complexes. (**A**) Schematic illustrating disassembly of the anti-parallel reporting L-SNARE-CN construct that was labeled with fluorescent dyes indicated by stars at residue 249 of syntaxin-1A and residue 28 of synaptobrevin-2. (**B**) Representative smFRET efficiency time traces of L-SNARE-CN. (**C**) Corresponding point-wise FRET efficiency histogram using all observed time traces. (**D**) Fraction of molecules without transitions (mean values ± SD, *Figure 9—source data 1*). (**E**) Probability distribution histogram of smFRET efficiency (black line, left axis) and the corresponding dwell times (red dots, right axis) obtained from the HMM. (**F–G**) smFRET efficiency dwell time histograms obtained from HMM. Right subpanels show the populations (mean values ± SD) of the dwell times in the short- and long-lived states (*Figure 9—source data 2*).

DOI: https://doi.org/10.7554/eLife.36497.032

The following source data and figure supplement are available for figure 9:

**Source data 1.** Data summary table for the results shown in *Figure 9D*.
DOI: https://doi.org/10.7554/eLife.36497.034
**Source data 2.** Data summary table for the results shown in *Figure 9F-G*.
DOI: https://doi.org/10.7554/eLife.36497.035
**Figure supplement 1.** A 7.5 M urea wash prevents improperly assembled L-SNARE complexes.
DOI: https://doi.org/10.7554/eLife.36497.033

assembled SNARE complex (e.g., with one or more of the SNARE motifs not being fully incorporated) that reverts to the disassembled state.

High ionic strength, low αSNAP concentration, mutation of charged residues of αSNAP involved in SNARE complex binding, or addition of Cpx all have similar effects. In each case, fewer SNARE complexes are disassembled, and the disassembly dwell times for SNARE complexes are shorter (*Figures 3–4* and *6*). We also found that the presence of the syntaxin-1A N-terminal Habc domain slowed both the disassembly and reassembly processes slightly (*Figure 5*), which may be explained by hindrance effects of the Habc domain on both formation of the 20S complex and on spontaneous reassembly of the SNARE complex.

## Regulation of NSF-mediated disassembly by Cpx

Cpx and αSNAP compete for binding to the SNARE complex (*Figure 6—figure supplement 1*). Previous reports differed on the effect of Cpx on SNARE complex disassembly in bulk (*Pabst et al., 2000*; *Winter et al., 2009*); in retrospect, these differing results can be explained by different αSNAP:Cpx molar ratios. At a ratio of ~18:1, Cpx had no effect (*Pabst et al., 2000*), whereas at a

ratio of ~3:1, Cpx had a substantial effect on NSF-mediated disassembly of the soluble SNARE complex (*Winter et al., 2009*). The presence of a membrane reduced the inhibitory effect by Cpx, presumably due to an effective local concentration increase of αSNAP as a result of its interaction with the membrane (*Winter et al., 2009*). All these results point to a critical role of αSNAP and its precise stoichiometry in the NSF-mediated disassembly activity, as discussed below.

A recent EM structure of the 20S complex allowed unique assignment of the positions of the individual SNARE components and their interactions with αSNAP molecules (*White et al., 2018*) revealing that two αSNAP molecules bind primarily to a portion of the surface formed by syntaxin-1A and synaptobrevin-2 in the ternary SNARE complex (*Figure 6G*). The EM structure of the L-20S complex reported here is similar to that of the recent EM structure of both the neuronal 20S complex (*White et al., 2018*) and the V7-20S complex (*Zhao et al., 2015*). Moreover, this intermolecular interaction has also been observed in another structure of the 20S complex that contained four αSNAP molecules (*Zhao et al., 2015*). While there is at present no evidence that a 20S complex cannot form with only one αSNAP molecule, all structures of 20S complex determined thus far contain a conserved structural motif consisting of two αSNAP molecules that interact with a portion of the surface formed by syntaxin-1A and synaptovrevin-2 in the ternary SNARE complex.

How might Cpx interfere with NSF-mediated disassembly considering the available structural information? Structurally, Cpx binds to the groove between syntaxin-1A and synaptobrevin-2 helices in the SNARE complex (*Chen et al., 2002*). Superposition of the structure of the αSNAP/L-SNARE subcomplex of the L-20S complex with that of the SNARE/Cpx complex suggests that Cpx competes with one of the two αSNAP molecules (*Figure 6G*), as corroborated by a competition assay (*Figure 6—figure supplement 1*). Because Cpx interferes with binding of only one of the two αSNAPs to the SNARE complex, this suggests that at least two αSNAPs are required for a key step of productive disassembly. If only one αSNAP is present, disassembly might not proceed to completion, as suggested by the presence of the low FRET efficiency spikes in our smFRET experiments (*Figures 2J* and *6F*). Moreover, a third αSNAP molecule may be engaged during the hydrolysis cycle (*Shah et al., 2015*).

How can disassembly occur in vivo in the presence of Cpx, which is such a strong inhibitor? In the neuron, there are approximately similar expression levels of Cpx, αSNAP, γSNAP, and to a lesser degree, βSNAP (*Clary et al., 1990*; *Nishiki et al., 2001*; *Söllner et al., 1993a*; *Whiteheart et al., 1992*; *Wilhelm et al., 2014*). At a 1:1 molar ratio of Cpx to αSNAP, there is little NSF-mediated disassembly of the SNARE complex in our smFRET experiments (see 10 μM Cpx, *Figure 6*). However, because the concentration of SNAPs in the neuron is roughly three times higher than that of Cpx, some disassembly would take place assuming similar affinities and activities of all SNAP isoforms. Moreover, membrane binding of αSNAP (*Winter et al., 2009*) might further increase the local SNAP concentration at the plasma membrane where *cis* SNARE complexes are present, thus mitigating a possible inhibitory effect of Cpx on NSF-mediated *cis* SNARE complex disassembly. On the other hand, sequestration of Cpx to the synaptic vesicle via its curvature sensing C-terminal domain (*Gong et al., 2016*) may increase the local Cpx concentration, effectively decreasing the local SNAP:Cpx molar ratio and leading to inhibition of unwanted NSF-mediated disassembly of *trans* SNARE complexes.

## NSF disassembles improperly assembled SNARE complexes

In addition to disassembling ternary SNARE complexes, NSF also disassembles incomplete or improperly formed SNARE complexes. For example, by using anti-parallel reporting FRET label pairs, we observed disassembly of improper SNARE complex configurations by NSF (*Figure 9*). Such anti-parallel configurations are often disassembled within one second. Interestingly, an atomic force microscopy study reported that the stability of an anti-parallel ternary SNARE complex is three-fold lower than the properly assembled parallel complex (*Liu et al., 2012*). We speculate that the fast disassembly rate observed for anti-parallel configurations is related to their lower stability.

Using the binary t-SNARE (syntaxin-1A/SNAP-25A) complex as an example, we also studied the disassembly of incomplete SNARE complexes. Our experiments revealed the presence of relatively long-lived intermediate states during NSF-mediated disassembly of the binary t-SNARE complex (*Figure 7*). Occasionally, the system reverted to the previous state instead of proceeding to full disassembly (*Figure 7B*). Similarly, we observed such intermediates during spontaneous reassembly. Although we did not observe intermediate states during NSF-mediated disassembly of the ternary

SNARE complex (*Figure 2*), they may occur on a faster timescale (i.e., faster than 100 msec) than is currently observable with our apparatus.

## Concluding remarks

Our single-molecule experiments revealed two temporal states (short- and long-lived) of SNARE complexes for both NSF-mediated disassembly and spontaneous reassembly that are influenced by various conditions, such as αSNAP and Cpx concentration. The long-lived states represent stable assembled and disassembled states of the SNARE complex, while the short-lived states represent incomplete or improper states. Future technical improvements may expand the accessible timescales of the assay, enabling further dissection of the short-lived transitions. Furthermore, our approach could be useful not only in the study of the effect of other accessory proteins in the disassembly process, but also in the study of the motions of NSF correlated with disassembly through analysis of intermolecular FRET between labeled SNARE complex and NSF. Together with the static structures from X-ray crystallography and cryo-EM, the dynamics extracted from these smFRET studies will ultimately provide a more complete understanding of the molecular mechanism of NSF-mediated disassembly of the SNARE complex, an essential process in the membrane fusion cycle.

## Materials and methods

### Proteins: plasmids, expression, and purification

The linked SNARE complex (L-SNARE) is composed of the cytoplasmic domain of rat syntaxin-1A (residues 181 to 262), mouse full-length SNAP-25A, and the cytoplasmic domain of rat synaptobrevin-2 (residue 25 to 96) linked by two synthetic linkers, Sp1 (GGSNRSGGSDSGGSGSQGSSGAGG SLNSKSSGGSGGSANSTASNSGQGSGGIEGLSGG) and Sp2 (SGGSGNRGGSDSGGSGGIEGLGG QSSGASGGLKNSGSGGSGGSNATSASNSGQGSSGGG), and terminated with an Avi-tag sequence (GSGGGSGGSGGLNDIFEAQKIEWHE) at the C-terminus of synaptobrevin-2 as previously described (*Zorman et al., 2014*). All native cysteines of the wildtype syntaxin-1A, SNAP-25A, and synaptobrevin-2 proteins were mutated to serine, and several combinations of cysteine mutations for fluorescent label sites at surface-exposed positions were introduced by using QuikChange (Agilent, Santa Clara, CA). (See also the section on Protein labeling.) The genes for the L-SNARE constructs were codon-optimized, synthesized along with a N-terminal tobacco etch virus (TEV) cleavable 6×-histidine tag, and inserted into a pJ414 vector (ATUM, Newark, CA).

L-SNARE expression and in vivo biotinylation were performed by co-expression into a BirA gene engineered pACYC184 *E. coli* (Avidity, Aurora, CO). Cells were grown in 1 L TB medium at 37°C until $OD_{600}$ reached about 0.8 and expression was induced with the addition of 0.5 mM IPTG in the presence of 0.1 mM biotin overnight at 25°C. The cell pellets were suspended in PBS buffer (50 mM $NaH_2PO_4$ pH 8.0, 300 mM NaCl, 0.5 mM DTT) supplemented with 0.5 mM PMSF and EDTA-free Complete Protease Inhibitor Cocktail tablets (Roche, Basel, Switzerland). The cells were lysed by sonication and the lysate was clarified by centrifugation with a JA-20 rotor (Beckman Coulter, Brea, CA) at 20,000 rpm for 30 min. The supernatant was nutated with 2 mL of Nickel-NTA agarose resin (Qiagen, Hilden, Germany) for 1 hr at 4°C. The resin was washed extensively with 40 mL PBS containing 10 mM imidazole and subsequently washed with 40 mL PBS buffer containing 7.5 M urea. Following the urea wash, the resin was re-equilibrated in 40 mM PBS and the protein was then eluted with PBS containing 400 mM imidazole and dialyzed into 20 mM Tris pH 8.0, 50 mM NaCl, 0.5 mM EDTA, and 0.5 mM DTT in the presence of 100 μg of TEV protease to remove the affinity tag. A final purification step was performed using HiTrap Q (GE Healthcare, Piscataway, NJ) anion exchange resin with a linear gradient of 50–600 mM NaCl in 20 mM Tris pH 7.5, and 0.5 mM TCEP. The purity of the protein was confirmed by SDS-PAGE.

For the EM studies, Chinese hamster NSF and rat αSNAP was expressed and purified as described before (*Zhao et al., 2015*). For all other studies, Chinese hamster NSF with a TEV cleavable N-terminal 6×-histidine tag was expressed from pPROEX-1 vector in *E. coli* BL21 (DE3) cells (Agilent, Santa Clara, CA) at 25°C overnight using autoinducing LB medium (*Studier, 2005*). After collecting the cells by centrifugation, they were resuspended in lysis buffer (50 mM Tris pH 8.0, 300 mM NaCl, 50 mM imidazole, 0.5 mM TCEP, 10% glycerol), and lysed by running through an Avestin C5 homogenizer 3 times at 15000 psi. The lysate was clarified by centrifugation and the supernatant

was bound to 5 mL of Nickel-NTA resin at 4°C, and further washed with the lysis buffer. NSF was eluted using elution buffer (lysis buffer supplemented with 350 mM imidazole). The eluate was pooled, concentrated, and supplemented with 1 mM EDTA and 1 mM ATP. The concentrated protein was immediately loaded onto a Superdex 200 16/60 column (GE Healthcare, Piscataway, NJ) pre-equilibrated with glycerol-free SEC Buffer (20 mM Tris pH 7.5, 100 mM NaCl, 1 mM EDTA, 1 mM ATP, 0.5 mM TCEP).

αSNAP with a TEV-cleavable N-terminal 10×-histidine tag was codon-optimized and cloned into the pJ414 vector (ATUM, Newark, CA), and expressed in *E. coli* BL21 (DE3) cells (Agilent, Santa Clara, CA) at 25°C overnight using autoinducing LB medium (*Studier, 2005*). The cell pellets were resuspended in lysis buffer (50 mM $NaH_2PO_4$ pH 8.0, 300 mM NaCl, 20 mM imidazole, and 0.5 mM TCEP), and lysed by running through an Avestin C5 homogenizer 3 times at 15000 psi. The lysate was clarified by centrifugation and the supernatant was bound to 5 mL of Nickel-NTA resin at 4°C and washed with lysis buffer. αSNAP was eluted in elution buffer (lysis buffer supplemented with 500 mM imidazole) and dialyzed into αSNAP buffer (20 mM Tris pH 8.0, 50 mM NaCl, 0.5 mM TECP) with 100 µg of TEV protease to remove the affinity tag. The protein was then bound to a Mono-Q HR 10/100 column (GE Healthcare, Piscataway, NJ) and eluted with a linear gradient of 50–500 mM NaCl in αSNAP buffer. Fractions containing pure αSNAP were pooled, concentrated, and injected onto a Superdex 200 16/60 column (GE Healthcare, Piscataway, NJ) pre-equilibrated with glycerol-free SEC Buffer (20 mM Tris pH 7.5, 100 mM NaCl, 0.5 mM TCEP).

Rat Cpx was cloned into the pTEV5 vector (*Rocco et al., 2008*) with a N-terminal TEV protease cleavable 6×-histidine tag. The protein was expressed and purified as previously described (*Choi et al., 2016*). Additionally, the purified protein was dialyzed in the same SEC Buffer (20 mM Tris pH 7.5, 100 mM NaCl, 0.5 mM TCEP) used in NSF and αSNAP purification.

For assembly of the L-20S complex (shown in *Figure 1* and *Figure 1—figure supplement 1*) and the 20S complex (shown in *Figure 1—figure supplement 2*), hexameric NSF loaded with ATP and EDTA was mixed with αSNAP and L-SNARE complex (i.e., with the linked SNARE complex) or with the unlinked SNARE complex at a molar ratio of 1:10:2 and incubated on ice for 30 min. The mixture was then concentrated and purified by size exclusion chromatography using a Superdex 200 10/300 column (GE Healthcare, Piscataway, NJ) pre-equilibrated with 50 mM Tris pH 8.0, 150 mM NaCl, 1 mM ATP, 1 mM EDTA, and 0.5 mM TCEP. The resulting peak fractions containing the L-20S complex or the 20S complex were pooled and concentrated to a final concentration of ~15 mg/mL for grid preparation for single particle cryo-EM images or a gel-based assay for protein quantification, respectively.

## Sample vitrification for single particle EM of the L-20S complex

The protein sample was supplemented with 0.05% (v/v) Nonidet P-40 before vitrification. Quantifoil Cu R1.2/1.3 grids (Quantifoil Micro Tools GmbH, Germany) were used directly without glow discharge. Plunge freezing was performed using an FEI Vitrobot (Thermo Fisher Scientific, Waltham, MA). Aliquots of 2.5 µL samples were applied on the grids followed by blotting for 3 to 4 s and plunging into liquid ethane.

## Cryo-EM data collection of the L-20S complex

Grids were transferred to FEI Titan Krios (Thermo Fisher Scientific, Waltham, MA) operated at 300 kV at the University of Michigan. Images were recorded on a Gatan K2 Summit direct electron detector operated in super-resolution counting mode following the established dose fractionation data acquisition protocol (*Li et al., 2013*). Images were recorded at a magnification corresponding to a calibrated pixel size of 1.01 Å. The dose rate on the detector was set to be ~10 electrons per pixel per second. The total exposure time was 8 s, leading to a total accumulated dose of 78 $e^-/Å^2$ on the specimen. Each image was fractionated into 40 frames, each with an accumulation time of 0.2 s. Dose-fractionated images were manually recorded using UCSFImage4 (written by Xueming Li). Defocus values ranged from −1.5 to −3.0 µm. Additional details are provided in *Table 1*.

## Image processing of the L-20S complex

Super-resolution counting images were 2 × 2 binned and motion corrected using MotionCorr (*Li et al., 2013*). Defocus values were determined for each micrograph using Gctf (*Zhang, 2016*).

Subsequent processing was performed using RELION 1.4 (*Scheres, 2012*). A map of the 20S complex from the previous study (*Zhao et al., 2015*) was used as the initial reference model. No symmetry was assumed throughout the entire process. Additional details are provided in *Figure 1—figure supplement 1*.

## Model building and refinement of the L-20S complex

To build a low-resolution model of the L-20S complex, a previous model of the 20S complex was used as a starting point (*Zhao et al., 2015*) but with two of the four αSNAP molecules omitted based on the observed EM density. Rigid body docking and minimization was performed using UCSF Chimera (*Pettersen et al., 2004*). The SNARE complex was replaced with the crystal structure of the original neuronal SNARE complex (PDB accession code: 1SFC) (*Sutton et al., 1998*). Regions from this initial model that did not have substantial density in the EM maps of the L-20S complex were removed, in particular at the N-terminal end of the SNARE complex where a higher resolution EM structure recently revealed an interaction with the pore of the D1 ring of NSF (*White et al., 2018*).

Molecular graphics and analyses were performed with either PyMOL (The PyMOL Molecular Graphics System, Version 1.7.0.5 Schrödinger, LLC.) or UCSF Chimera. Chimera is developed by the Resource for Biocomputing, Visualization, and Informatics at the University of California, San Francisco (supported by NIGMS P41-GM103311).

## Protein labeling

Cysteine mutations were introduced into the L-SNARE construct at surface exposed positions by site-directed mutagenesis using Quik-Change Kit (Agilent, Santa Clara, CA). The cysteine residues were stochastically labeled with the fluorescent dyes Alexa 555 $C_2$ and/or Alexa 647 $C_2$ maleimide (Thermo Fisher Scientific, Waltham, MA) in 20 mM Tris pH 7.5, 300 mM NaCl, and 0.5 mM TCEP overnight at 4°C. The labeled proteins and the excess free dyes were separated on a homemade column packed with Sephadex G50 resin (GE Healthcare, Piscataway, NJ) and equilibrated with 20 mM Tris pH 7.5 and 100 mM NaCl.

As the cysteine residues of αSNAP are critical for folding, an unnatural amino acid (UAA) labeling strategy was used for the experiments shown in *Figure 6—figure supplement 1*. Site-specific genetic encoding of a UAA by amber suppression allows the incorporation of biorthogonal chemical functional groups anywhere into the protein, preserving the native cysteine residues (*Chin et al., 2002*; *Tyagi and Lemke, 2013*). Residue 12 of αSNAP was mutated to the amber stop codon TAG with the termination stop codon as TAA in a pTEV5 vector with a N-terminal TEV-cleavable 6×-histidine tag. Co-expression of mutant αSNAP and the suppressor tRNA and aminoacyl-tRNA synthetase for incorporation of (L)−4-Azido-methyl-phenylalanine (SynChem, Elk Grove Village, IL) was performed from the pEVOL plasmid obtained from Peter Schulz (The Scripps Research Institute, San Diego, CA) in one liter TB medium supplemented with 50 µg/ml ampicillin (Sigma-Aldrich, St. Louis, MO) and 34 µg/ml chloramphenicol (Affymetrix, Cleveland, OH) at 37°C. When the $OD_{600}$ reached 0.6, 2 mM (L)−4-Azido-methyl-phenylalanine was added. When the $OD_{600}$ reached 0.8, αSNAP expression was induced with the addition of 0.5 mM IPTG while the unnatural amino acid machinery was induced with the addition of 0.02% arabinose (Sigma-Aldrich, St. Louis, MO). Expression was performed overnight at 25°C. Purification was conducted using the same protocol for unlabeled αSNAP as described above. The 33 kDa full-length αSNAP was confirmed by SDS-PAGE showing successful incorporation of the UAA.

The labeling of αSNAP with a fluorescent dye was conducted using copper-free click chemistry (*Tyagi and Lemke, 2013*). Copper-free click chemistry was performed by the azide/DIBO reaction, where the azide-modified group of the unnatural amino acid reacts to the fluorescent dyes containing DIBO derivatives in the absence of any catalysts (*Ning et al., 2008*). αSNAP labeling was carried out in the presence of a five molar excess of DIBO-Alexa 555 (Thermo Fisher Scientific, Waltham, MA) in 20 mM Tris pH 7.5, 300 mM NaCl, and 0.5 mM TCEP overnight at 4°C. The free dye was removed using a column packed with Sephadex G50 resin (GE Healthcare, Piscataway, NJ) and equilibrated in 20 mM Tris pH 7.5, 100 mM NaCl, and 0.5 mM TCEP.

## smFRET assay of NSF-mediated disassembly of SNARE complexes

All proteins were dialyzed in the same final buffer with the same ionic strength (100 mM NaCl), except where stated otherwise. For all smFRET experiments, the surface inside the chamber of the microscope slide was passivated by 10 mg/ml egg phosphatidylcholine 50 nm liposomes (Avanti Polar lipids, Alabaster, AL) coated with 1 mg/ml biotinylated BSA to mimic the lipid environment inside the cell (*Choi et al., 2012*). 0.1 mg/ml streptavidin was added to surface-tether the biotinylated and fluorescently labeled L-SNARE complexes. Tethering was performed with labeled L-SNARE samples diluted to about 50 pM in TBS (20 mM Tris pH 7.5 and 100 mM NaCl) to produce a well separated density of about 500 molecules per $45 \times 90\ \mu m^2$ field of view. To begin monitoring the NSF-mediated disassembly and the spontaneous reassembly of the surface-tethered L-SNARE complexes, 1 μM NSF, 10 μM αSNAP, 1 mM ATP, and 1 mM $MgCl_2$ were added to the observation buffer (consisting of TBS with 1% glucose in the presence of oxygen scavenger (20 units/ml glucose oxidase, 1000 units/ml catalase) and triplet-state quencher (100 μM cyclooctatetraene) to prevent fast photobleaching and blinking of the dye molecules).

## Single-molecule competition assay

For the experiments shown in *Figure 6—figure supplement 1*, biotinylated and Alexa 647 labeled L-SNARE complex at residue 193 of syntaxin-1A was surface-tethered to polyethylene glycol (PEG) coated surface consisting of 0.1% (w/v) biotinylated-PEG using streptavidin as a linkage. The surface density of L-SNAREs was increased to about 2000 molecules per field of view to optimize the competition effect of Cpx and αSNAP on binding with the surface-tethered SNARE complex. 10 nM of αSNAP labeled with Alexa 555 at residue 12 was concurrently incubated with varying concentrations of Cpx inside the chamber of the microscope slide for 5 min before rinsing away unbound proteins with the observation buffer to count the number of Alexa 555 labeled αSNAP bound to the SNARE complex. For each condition, three independent fields of view were observed using direct excitation with 635 nm and 532 nm laser illumination, respectively.

The counts of Alexa 555 labeled αSNAP were normalized by dividing the number of αSNAP molecules by the number of surface-tethered L-SNARE complexes.

## Data collection and analysis of the single-molecule experiments

Details related to instrumental setup, data collection, and analysis have been described elsewhere (*Lai et al., 2017*). Briefly, imaging data were collected at a frame rate of 10 Hz for about 200 s using the smCamera program kindly provided by Taekjip Ha (Johns Hopkins University, Baltimore, MD) with 532 nm laser illumination (donor excitation) until most of the dyes photobleached.

The data were processed using MATLAB (Mathworks, Natick, MA) and Igor Pro scripts (WaveMetrics, Portland, OR). The FRET efficiency E was calculated by

$$E = \frac{I_A}{(I_A + I_D)}$$

where $I_A$ and $I_D$ are the fluorescence intensities of leakage corrected acceptor and donor emissions (*McCann et al., 2010*).

For *Figures 2–5,7–9*, and *Figure 2—figure supplement 1* the emission pathway contained a 640 nm dichroic mirror (Semrock, Rochester, NY), and a 405/488/532/635 quad-notch filter, where the leakage of donor fluorescence into the acceptor channel was 1.7% and the leakage of acceptor fluorescence into the donor channel was 16.5% (*Lai et al., 2017*). For *Figure 6*, *Figure 6—figure supplement 1*, and *Figure 7—figure supplement 1*, and *Figure 9—figure supplement 1*, the emission pathway was replaced with an OptoSplit II image splitter (Andor Technology, South Windsor, CT) containing a 552 nm LP filter (Semrock, Rochester, NY), a 635 nm dichroic mirror (Semrock, Rochester, NY), a 582/64 BP filter (Semrock, Rochester, NY), and a 698/70 BP filter (Semrock, Rochester, NY) where no leakage between donor and acceptor channels was present.

The fields of view were analyzed with the MATLAB script in order to detect all fluorescent spots with donor dye emission (referred to as all *molecules*), and for each molecule that also showed acceptor dye emission, the FRET efficiency was calculated. For molecules with multiple FRET transitions (defined as molecules with transitions), the FRET efficiency time traces were divided into three sections (I, II, III) where section I is the initial FRET efficiency state, section III is the final high FRET

efficiency state, and section II comprises of all FRET efficiency states in between I and III (*Figure 2—figure supplement 1*). Sections I and III were excluded from analysis because the actual dwell times are unknown; that is, the start time of the initial state is unknown for section I, and for section III the end time is unknown since the final step in the FRET efficiency time trace can be due to acceptor photobleaching (*Figure 2—figure supplement 1A*) or donor photobleaching (*Figure 2—figure supplement 1B*). We define 'molecules without transitions' as molecules that exhibit constant high FRET efficiency until the end of the observation period or until early photobleaching of the acceptor dye (*Figure 2B*). The fraction of molecules without transitions was calculated as the number of molecules without transitions divided by the total number of molecules. Molecules that did not show co-localized acceptor fluorescence intensity (*e.g.*, molecules labeled with only one or two donor dyes), or that suggested the presence of multiple complexes in one spot were not included in the counts of the molecules with and without transitions. Moreover, the fraction of molecules without transitions can never reach more than 50% due to stochastic labeling, incomplete labeling efficiency, and other experimental considerations.

The process was repeated for the specified number of fields of view (at least three, see Source Data files) using the same protein and surface preparations. Where specified, standard deviations were calculated from these individual experiments.

## Single-molecule FRET transition analysis using hidden Markov modeling

To extract the FRET efficiency states and the corresponding dwell times of FRET efficiency states, we analyzed the smFRET efficiency traces using the HMM software HaMMy (version 4.0) (*McKinney et al., 2006*). Each molecule was analyzed separately. For all two-state cases (*Figures 2–6,8,9*) initial guesses of FRET efficiencies of $E = 0.2$ and $E = 0.8$ were used (based on the observed point-wise FRET distributions, e.g., *Figure 2C*). The HMM algorithm then refines these estimates to fit the data using a maximum likelihood approach. The probability distribution histogram of the fitted FRET efficiencies from HHM (*Figure 2F*) was used to define the FRET efficiency states, with low FRET efficiency $0 < E \leq 0.5$ and high FRET efficiency $0.5 < E \leq 1$. The dwell times for each of the two states were plotted in separate histograms on both linear and logarithmic timescales. We used the local minimum in the logarithmic plots in *Figure 2G–H* to partition the histograms into short-lived and long-lived states (0.56 s in *Figure 2G*, and 0.32 s in *Figure 2H*), and calculated the areas under the respective regions to determine the populations of these states. Since we measured the largest data set for L-SNARE-CC molecules (*Figure 2*), and in order to make the analysis comparable between different conditions and constructs, we used the same partition for short- and long-lived states for the dwell time histograms for all the other two-state cases (*Figures 3–6, 8–9*).

For the three-state case (*Figure 7*), initial guesses of $E = 0.2$, $E = 0.5$, and $E = 0.8$ were used for HMM (again, based on the observed point-wise FRET distributions, e.g., *Figure 7C,D*). The probability distribution histograms of the fitted FRET efficiency from HHM (*Figure 7F,G*) were used to define the FRET efficiency states. For L-SNARE$_{binary}$-CC1: low FRET $0 < E \leq 0.45$, mid FRET $0.45 < E \leq 0.8$, and high FRET $0.8 < E \leq 1$. For L-SNARE$_{binary}$-CC2: low FRET $0 < E \leq 0.35$, mid FRET $0.35 < E \leq 0.7$, and high FRET $0.7 < E \leq 1$. The corresponding dwell time histograms for the three FRET efficiency states were plotted and partitioned into short- and long-lived dwell times (*Figure 7H,I*).

## Acknowledgements

We thank William Weis and Thomas Südhof for discussions, Georgios Skiniotis for cryo-EM data collection at the University of Michigan, and Yunxiang Zhang for help with the IDL software for the analysis of single-molecule fluorescence data. We acknowledge support by the National Institutes of Health (R37MH63105 to ATB) and by a postdoctoral fellowship from the Helen Hay Whitney Foundation supported by the Howard Hughes Medical Institute awarded to KIW.

## Additional information

### Competing interests

Axel T Brunger: Reviewing editor, *eLife*. The other authors declare that no competing interests exist.

## Funding

| Funder | Grant reference number | Author |
|---|---|---|
| National Institutes of Health | R37MH63105 | Axel T Brunger |
| Howard Hughes Medical Institute | | Axel T Brunger |

The funders had no role in study design, data collection and interpretation, or the decision to submit the work for publication.

## Author contributions

Ucheor B Choi, Conceptualization, Formal analysis, Investigation, Methodology, Writing—original draft, Writing—review and editing; Minglei Zhao, Conceptualization, Formal analysis, Investigation, Writing—original draft, Writing—review and editing; K Ian White, Conceptualization, Validation, Writing—original draft, Writing—review and editing; Richard A Pfuetzner, Data curation, Writing—original draft; Luis Esquivies, Resources, Writing—original draft; Qiangjun Zhou, Conceptualization, Resources; Axel T Brunger, Conceptualization, Supervision, Funding acquisition, Writing—original draft, Project administration, Writing—review and editing

## Author ORCIDs

Ucheor B Choi (ID) http://orcid.org/0000-0003-1541-2967
Minglei Zhao (ID) https://orcid.org/0000-0001-5832-6060
K Ian White (ID) http://orcid.org/0000-0001-8182-3655
Richard A Pfuetzner (ID) http://orcid.org/0000-0002-1741-2330
Luis Esquivies (ID) http://orcid.org/0000-0002-1750-6775
Qiangjun Zhou (ID) http://orcid.org/0000-0003-1789-9588
Axel T Brunger (ID) http://orcid.org/0000-0001-5121-2036

## Decision letter and Author response

Decision letter https://doi.org/10.7554/eLife.36497.040
Author response https://doi.org/10.7554/eLife.36497.041

# Additional files

## Supplementary files

• Transparent reporting form
DOI: https://doi.org/10.7554/eLife.36497.036

## Data availability

The EM map associated with this paper has been deposited in the wwPDB under the accession number EMD-8944.

The following dataset was generated:

| Author(s) | Year | Dataset title | Dataset URL | Database, license, and accessibility information |
|---|---|---|---|---|
| Brunger A, Zhao M | 2018 | 20S supercomplex consisting of linked neuronal SNARE complex, alpha-SNAP, and N-ethylmaleimide sensitive factor (NSF) | http://emsearch.rutgers.edu/atlas/8944_summary.html | Publicly available at the RCSB Protein Data Bank (accession no: EMD-8944) |

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
