## [Decision Letter]

Thank you for submitting your article "NSF-mediated disassembly of on and off-pathway SNARE complexes, and inhibition by complexin" for consideration by *eLife*. Your article has been reviewed by three peer reviewers, a Reviewing Editor, and by Vivek Malhotra as the Senior Editor and Reviewer #2. The following individual involved in review of your submission has agreed to reveal their identity: Yongli Zhang (Reviewer #1).

Your manuscript has been evaluated by three experts and after extensive consultation, we have decided to invite a revision that addresses the following major issues. Most of the issues require rewriting the text to highlights the caveats and means to address them experimentally, if possible.

Essential revisions:

1) Although the assay provides new insight into the life time of the assembled and disassembled states of SNARE proteins under NSF action, the paper often omits a molecular explanation for the individual states and their transition events or draws conclusions that are not directly supported by the presented data. For example, it remains unexplained why αSNAP affects not only the disassembly but also the kinetics of the assembly process (Figure 3I,J). This issue should be clarified and discussed. Is there an experiment that can address this issue? If not, then please clarify?

2) Furthermore, the authors show that addition of complexin specifically shortens the lifetime of the low FRET efficiency state without changing the high FRET dwell time. The latter one reflects the life time of the assembled SNARE complex. Yet, the authors conclude that complexin inhibits the NSF-mediated disassembly kinetics of the SNARE complex, which seems to be at odds to the presented data. They furthermore suggest that complexin action may lead to incomplete SNARE disassembly but provide no evidence for states of intermediate FRET efficiency as observed for binary complex disassembly (e.g. Figure 7). Indeed, the results rather favour the view that complexin specifically facilitates the assembly of ternary SNARE complexes without changing disassembly kinetics. Please explain.

3) In this context, the exemplary trace shown in Figure 6B (10 µM Cpx WT) shows a high FRET state that is frequently interrupted by spike-like low FRET deflections without having an impact on the frequency distribution of the high FRET dwell time (Figure 6E). Here, it is also important to note that the manuscript provides no information regarding important constraints used in the analysis like thresholding of the transition states, signal-to-noise ratio etc. Please explain.

4) In the context of the observed competition between complexin and the SNAP variants for binding to the ternary SNARE complex, the authors suggest that curvature sensing C-terminal domain of complexin may sequester the protein to vesicles, thus effectively increasing the SNAP:complexin molar ratio – a favorable scenario for SNARE complex disassembly. However, previous studies have shown that vesicle association of complexin increases its 'effective' concentration at the site of action (Wragg et al., 2013) and that the vesicular localization of the protein is a prerequisite for its inhibitory action (e.g. Gong et al., 2016). Thus, it is not clear why the authors now propose that it is membrane binding of complexin that facilitates αSNAP-mediated SNARE disassembly and fusion. Please clarify this point.

5) In their manuscript the authors consistently cite White et al., 2018, manuscript in preparation, even for some of their major conclusions. While it is often helpful to mention corroborating evidences to strengthen conclusions drawn in the paper, it is impossible for reviewers or readers to determine the significance of this citation as it is yet to be published. Please limit the citation of this unpublished work.

6) The manuscript is entirely devoid of any statistical analysis. Furthermore, it remains often unclear whether averages were determined over the entire number of life time states, recording epochs or over the number of independent experiments, the latter would be most valid. For normally distributed parameters the naturally occurring scatter (determined as SD/mean=coefficient of variation) increases with increasing amplitude of the signal, which is not found in Figure 3C, H. Please explain this phenomenon. New statistics should be presented in the corresponding figure legends or the main body of the text.

7) Short-lived low-FRET states are frequently observed under many experimental conditions: at high NaCl concentrations, at low αSNAP concentrations, and in the presence of mutant αSNAP or 10 μm complexin. The authors interpreted these states to be partially disassembled SNARE states or failed disassembly. Although this interpretation is reasonable, an alternative explanation is possible. Are these short-lived states simply caused by fast SNARE reassembly upon full SNARE disassembly? Please explain this.

8) The authors speculated that, at a low αSNAP concentration, "the unproductive low FRET efficiency spikes may be a consequence of an only partially occupied second α-SNAP site" (subsection “Lowering the αSNAP concentration decreases the disassembly activity”). Are the spikes caused by binding of only one αSNAP? The authors have clearly shown that SNARE complexes contain multiple binding sites for αSNAP. αSNAP binding to different sites and different binding stoichiometry may lead to different disassembly pathways and kinetics. Please rewrite the text to better explain the data with caveats that help will resolve this issue.

9) Regarding t-SNARE disassembly, Figure 7B shows fast t-SNARE conformational transitions. It is not obvious how much NSF contributes to these transitions, because t-SNARE complexes alone exhibits fast transitions (Weninger et al., 2008). It may be useful to compare the FRET traces with and without NSF-dependent disassembly activity. In addition, NSF does not seem to significantly increase the populations of the low FRET states (the unfolded t-SNARE states), especially for the CC2 construct (compare Figure 7E and Figure 7—figure supplement 1C). In contrast, the intermediate state appears to be a distinct state caused by NSF. Does NSF only partially disassemble the t-SNARE complex? Can this be tested experimentally?

10) The SNARE disassembly assay was carried out at the membrane surface. It is known that αSNAP binds membranes, which can greatly accelerate the rate of SNARE disassembly by NSF (Winter et al., 2009). However, the Discussion section seems to suggest that membrane binding does not play an important role in this assay. Does membrane binding contribute to the NSF activity reported in this work? If membrane binding is important, it may be illustrated in some of the diagrams, such as Figure 6A. Please address this by better explanation and redoing Figure 6A.

11) The authors showed evidence that two αSNAP molecules bind to the interface between syntaxin and VAMP2, which is very interesting. The evidence is consistent with recent observations that αSNAP stabilizes zippered SNARE complex (Ma et al., 2016) and Sec17 promotes membrane fusion (Zick et al., 2015). However, the fact that NSF disassembles all SNARE complexes, mis-assembled SNARE complexes, and t-SNARE complexes (with a broken interface) indicates certain plasticity in binding between αSNAP and SNARE complexes, including binding stoichiometry and site (Vivona et al., 2013). Please discuss this feature.

Reviewer #1:

This is one of the most interesting papers I have ever read on the kinetics of SNARE disassembly by NSF. In this paper, Brunger et al. reported a new assay to accurately detect dynamic SNARE disassembly by NSF and spontaneous SNARE reassembly. Using a SNARE construct that joins three neuronal SNARE proteins into one polypeptide, they could detect repetitive NSF-dependent disassembly of a single SNARE complex by single-molecule fluorescence energy transfer (smFRET). This assay revealed many important findings: (1) Two αSNAP molecules binds to each SNARE complex, as opposed to four αSNAP molecules previous reported for a SNARE complex containing SNAP-25 without its linker region. (2) High NaCl and low αSNAP concentrations reduce the disassembly rate. (3) Complexin attenuates SNARE disassembly. (4) NSF efficiently disassembles t-SNARE complexes and mis-assembled SNARE complexes. The data are elegant, well presented, clearly described, and support the main conclusions.

Reviewer #2:

The authors present a quantitative assay to measure dynamics of SNARE assembly and disassembly. The data are of high quality and important for a comprehensive understanding of the SNARE mediated membrane fusion. The authors findings clarify an important issue pertaining to the function of complexins in SNARE assembly reaction. The authors data show nicely that complexins compete with one of the two αa-SNAP molecules in the SNARE complex and compete with NSF mediated disassembly.

The findings are relatively straightforward and merit publication in *eLife*.

Reviewer #3:

The manuscript by Choi et al. addresses the role of the ATPase NSF in ternary and binary SNARE complex disassembly, using single molecule fluorescence resonance energy transfer (smFRET) experiments. The authors have devised a new experimental set up to observe multiple rounds of SNARE assembly and their NSF-mediated disassembly by covalently linking the individual SNARE proteins with flexible linkers. By this, the manuscript lines up in a row of important contributions over the last years studying the mechanism of NSF mediated SNARE complex recycling (Ryu et al., 2015; Zhao et al., 2015).

The manuscript by Choi et al., shows that disassembly of single ternary SNARE complexes is sensitive to ionic strength of the milieu, concentration of the NSF co-factor αSNAP and to mutations in αSNAP that mask its electrostatic interaction with negative surface charges on the SNARE complex. (See also Zhao et al., 2015.)

Furthermore, the authors report that complexin 1 decreases the disassembly activity of NSF in a concentration dependent manner, with very short-lived low FRET spikes (reflecting the dissociated state of the SNARE proteins) at high complexin 1 concentration (10µM). Comparing the superimposed structures of α-SNAP/SNARE complex with those of complexin/SNARE complexes the authors posited that complexin may compete with αSNAP for SNARE complex interaction.

While the majority of experiments are well done, I still have substantial concerns that should be addressed before consideration for publication.

1) Although the assay provides new insight into the life time of the assembled and disassembled states of SNARE proteins under NSF action, the paper often omits a molecular explanation for the individual states and their transition events or draws conclusions that are not directly supported by the presented data. For example, it remains unexplained why αSNAP affects not only the disassembly but also the kinetics of the assembly process (Figure 3I,J). This issue should be clarified and discussed.

Furthermore, the authors show that addition of complexin specifically shortens the lifetime of the low FRET efficiency state without changing the high FRET dwell time. The latter one reflects the life time of the assembled SNARE complex. Yet, the authors conclude that complexin inhibits the NSF-mediated disassembly kinetics of the SNARE complex, which seems to be at odds to the presented data. They furthermore suggest that complexin action may lead to incomplete SNARE disassembly but provide no evidence for states of intermediate FRET efficiency as observed for binary complex disassembly (e.g. Figure 7). Indeed, the results rather favour the view that complexin specifically facilitates the assembly of ternary SNARE complexes without changing disassembly kinetics.

In this context, the exemplary trace shown in Figure 6B (10 µM Cpx WT) shows a high FRET state that is frequently interrupted by spike-like low FRET deflections without having an impact on the frequency distribution of the high FRET dwell time (Figure 6E). Here, it is also important to note that the manuscript provides no information regarding important constraints used in the analysis like thresholding of the transition states, signal-to-noise ratio etc.

2) In the context of the observed competition between complexin and the SNAP variants for binding to the ternary SNARE complex, the authors suggest that curvature sensing C-terminal domain of complexin may sequester the protein to vesicles, thus effectively increasing the SNAP:complexin molar ratio – a favorable scenario for SNARE complex disassembly. However, previous studies have shown that vesicle association of complexin increases its 'effective' concentration at the site of action (Wragg et al., 2013) and that the vesicular localization of the protein is a prerequisite for its inhibitory action (e.g. Gong et al., 2016). Thus, it is not clear why the authors now propose that it is membrane binding of complexin that facilitates αSNAP-mediated SNARE disassembly and fusion. The authors should clarify this point.

3) In their manuscript the authors consistently cite White et al., 2018, manuscript in preparation, even for some of their major conclusions. While it is often helpful to mention corroborating evidences to strengthen conclusions drawn in the paper, it is impossible for reviewers or readers to determine the significance of this citation as it is yet to be published. Thus, limiting the citation of this unpublished work might be prudent.

4) The manuscript is entirely devoid of any statistical analysis. Furthermore, it remains often unclear whether averages were determined over the entire number of life time states, recording epochs or over the number of independent experiments, the latter would be most valid. For normally distributed parameters the naturally occurring scatter (determined as SD/mean=coefficient of variation) increases with increasing amplitude of the signal, which is not found in Figure 3.C, H. The authors should explain this phenomenon. New statistics should be presented in the corresponding figure legends or the main body of the text.

---

## [Author Response]

Essential revisions:
*1) Although the assay provides new insight into the life time of the assembled and disassembled states of SNARE proteins under NSF action, the paper often omits a molecular explanation for the individual states and their transition events or draws conclusions that are not directly supported by the presented data. For example, it remains unexplained why* αSNAP affects not only the disassembly but also the kinetics of the assembly process (Figure 3I,J). This issue should be clarified and discussed.Is there an experiment that can address this issue? If not, then please clarify?

We thank the reviewers for this comment, which prompted us to reanalyze our data using a hidden Markov model (HMM), a commonly-used statistical approach to analyze single molecule FRET efficiency data and transitions between states. The particular method (McKinney et al.) interprets the observed point-wise (i.e., time-binned) FRET efficiency traces in terms of as a hidden Markov process with the specified number of states. This method produces the most likely FRET efficiency values (based on a maximum likelihood approach) for these states for each individual molecule, and also determines their interconversion rates and dwell times. The apparent increase in low-FRET dwell time that we reported in our original submission is due to a redistribution of long-lived (i.e., lasting many seconds) and short-lived (lasting ≲ sec) low-FRET states. The HMM analysis is described in more detail in the revised manuscript.

Additionally, we analyzed the number of SNARE complexes that do not undergo any disassembly event during their respective observation periods.

At low αSNAP concentration, the fraction of molecules that do not undergo any disassembly increases, and mostly short-lived states are observed, whereas at higher αSNAP concentration, many long-lived low-FRET (i.e., disassembled) states are observed. We interpret the short-lived states as failed disassembly attempts or assembly with immediate reassembly. At a low level of αSNAP concentration, the assembly rate of the SNARE complex is difficult to measure because few instances of long-lived, low-FRET efficiency exist, thereby preventing assessment of αSNAP on SNARE complex assembly. In other words, the convolution of the non-productive disassembly spikes as well as the full disassembly events in the previous version of the manuscript gave the false impression of an increase in the reassembly rate. We have revised the text accordingly.

Possible molecular explanations for our observed single molecule results have been added to the Discussion section.

2) Furthermore, the authors show that addition of complexin specifically shortens the lifetime of the low FRET efficiency state without changing the high FRET dwell time. The latter one reflects the life time of the assembled SNARE complex. Yet, the authors conclude that complexin inhibits the NSF-mediated disassembly kinetics of the SNARE complex, which seems to be at odds to the presented data. They furthermore suggest that complexin action may lead to incomplete SNARE disassembly but provide no evidence for states of intermediate FRET efficiency as observed for binary complex disassembly (e.g. Figure 7). Indeed, the results rather favour the view that complexin specifically facilitates the assembly of ternary SNARE complexes without changing disassembly kinetics. Please explain.

We thank the reviewers for the comment. As mentioned above, we reanalyzed our data with HMM and also analyzed the fraction of SNARE complex molecules that do not undergo any disassembly events. The analysis shows that at higher complexin concentration, the fraction of molecules without disassembly events increases, and the short-lived, low-FRET efficiency states dominate over long-lived low-FRET efficiency states, (i.e., complexin has a similar effect as lowering the αSNAP concentration). While we cannot rule out an additional effect of complexin on SNARE complex reassembly, the competition experiment (Figure 6 – —figure supplement 1) clearly shows that complexin displaces αSNAP, which in turn will lower the efficiency of NSF-mediated SNARE complex disassembly. These results explain the apparent increase in reassembly rate in the previous version of the manuscript, as discussed in the previous point. We have revised the text accordingly.

3) In this context, the exemplary trace shown in Figure 6B (10 µM Cpx WT) shows a high FRET state that is frequently interrupted by spike-like low FRET deflections without having an impact on the frequency distribution of the high FRET dwell time (Figure 6E). Here, it is also important to note that the manuscript provides no information regarding important constraints used in the analysis like thresholding of the transition states, signal-to-noise ratio etc. Please explain.

As mentioned above, we have reanalyzed the data with HMM. We made initial guesses for FRET efficiency states for HMM by inspecting the point-wise FRET histograms. The algorithm then refines these estimates to fit the data using a maximum likelihood approach. Additionally, HMM was used to distinguish between processes with short- and long-lived dwell times of instances of high and low FRET efficiency. We determined the partition between the short- and long-lived processes by identifying local minima of the dwell-time distributions. These were best visualized by plotting the distributions on a logarithmic time scale

We have computed standard deviations as specified in the Materials and methods section using multiple independent measurements of the same sample and surface preparations.

*4) In the context of the observed competition between complexin and the SNAP variants for binding to the ternary SNARE complex, the authors suggest that curvature sensing C-terminal domain of complexin may sequester the protein to vesicles, thus effectively increasing the SNAP:complexin molar ratio – a favorable scenario for SNARE complex disassembly. However, previous studies have shown that vesicle association of complexin increases its 'effective' concentration at the site of action (Wragg et al., 2013) and that the vesicular localization of the protein is a prerequisite for its inhibitory action (e.g. Gong et al., 2016). Thus, it is not clear why the authors now propose that it is membrane binding of complexin that facilitates* α*SNAP-mediated SNARE disassembly and fusion. Please clarify this point.*

We apologize for the misunderstanding—we actually meant the opposite. We suggest that the αSNAP:complexin molar ratio is lowered by localization of complexin molecules to synaptic vesicles via the membrane curvature of the synaptic vesicle membrane. This effective increase in complexin concentration at synaptic vesicle docking sites would effectively prevent NSF-mediated disassembly of *trans* SNARE complexes. This actually would be desirable, as it would prevent unwanted *trans* SNARE complex disassembly prior to fusion. Of course, in addition to complexin, other factors may also cooperate in preventing disassembly of *trans* SNARE complexes. We have revised the text accordingly.

5) In their manuscript the authors consistently cite White et al., 2018, manuscript in preparation, even for some of their major conclusion. While it is often helpful to mention corroborating evidences to strengthen conclusions drawn in the paper, it is impossible for reviewers or readers to determine the significance of this citation as it is yet to be published. Please limit the citation of this unpublished work.

Meanwhile, this work has been also submitted to *eLife,* where it is currently undergoing in-depth review. We have uploaded this submitted manuscript as a related manuscript file.

6) The manuscript is entirely devoid of any statistical analysis. Furthermore, it remains often unclear whether averages were determined over the entire number of life time states, recording epochs or over the number of independent experiments, the latter would be most valid. For normally distributed parameters the naturally occurring scatter (determined as SD/mean=coefficient of variation) increases with increasing amplitude of the signal, which is not found in Figure 3.C, H. Please explain this phenomenon. New statistics should be presented in the corresponding figure legends or the main body of the text.

As mentioned above, we have reanalyzed our data using HMM. Moreover, as specified in the Materials and methods section, we have also computed standard deviations for many quantities (such as population averages and rates) using multiple independent measurements of the same sample and surface preparations.

7) Short-lived low-FRET states are frequently observed under many experimental conditions: at high NaCl concentrations, at low αSNAP concentrations, and in the presence of mutant αSNAP or 10 μm complexin. The authors interpreted these states to be partially disassembled SNARE states or failed disassembly. Although this interpretation is reasonable, an alternative explanation is possible. Are these short-lived states simply caused by fast SNARE reassembly upon full SNARE disassembly? Please explain this.

It is possible that the short spikes are complete disassembly followed by immediate reassembly. Because we cannot distinguish between these possibilities with our current setup, we now offer this as an alternative explanation in the text.

*8) The authors speculated that, at a low αSNAP concentration, "the unproductive low FRET efficiency spikes may be a consequence of an only partially occupied second αSNAP site" (subsection “Lowering the αSNAP concentration decreases the disassembly activity”). Are the spikes caused by binding of only one αSNAP? The authors have clearly shown that SNARE complexes contain multiple binding sites for αSNAP.* α*SNAP binding to different sites and different binding stoichiometry may lead to different disassembly pathways and kinetics. Please rewrite the text to better explain the data with caveats that help will resolve this issue.*

We thank the reviewers for the comment. Our EM structures suggests that between 2 and 4 αSNAPs are bound in the initial state (Zhao et al., 2015; White, et al., 2018). Even if initially only two αSNAPs are bound in the 20S complex, an additional αSNAP molecule may be engaged during the disassembly process (Shah et al., 2015). It is possible, or even likely, that the short-lived low-FRET states at low αSNAP concentration correspond to the presence of one αSNAP, which would be insufficient for complete disassembly. We have revised the text accordingly.

9) Regarding t-SNARE disassembly, Figure 7B shows fast t-SNARE conformational transitions. It is not obvious how much NSF contributes to these transitions, because t-SNARE complexes alone exhibits fast transitions (Weninger et al., 2008). It may be useful to compare the FRET traces with and without NSF-dependent disassembly activity. In addition, NSF does not seem to significantly increase the populations of the low FRET states (the unfolded t-SNARE states), especially for the CC2 construct (compare Figure 7E and Figure 7—figure supplement 1C). In contrast, the intermediate state appears to be a distinct state caused by NSF. Does NSF only partially disassemble the t-SNARE complex? Can this be tested experimentally?

We thank the reviewers for the comment. Actually, there are no transitions between the states of the binary complex in the absence of NSF on the time scale of our experiments. We have added representative traces to Figure 7—figure supplement 1B. We also now explicitly mention in the text that the transitions that we observe in Figure 7 are induced by NSF/αSNAP.

10) The SNARE disassembly assay was carried out at the membrane surface. It is known that αSNAP binds membranes, which can greatly accelerate the rate of SNARE disassembly by NSF (Winter et al., 2009). However, the discussion at the bottom of page 21 seems to suggest that membrane binding does not play an important role in this assay. Does membrane binding contribute to the NSF activity reported in this work? If membrane binding is important, it may be illustrated in some of the diagrams, such as Figure 6A. Please address this by better explanation and redoing Figure 6A.

We apologize for the misunderstanding. We do not intend to suggest that membrane binding does not play a role in NSF/αSNAP mediated SNARE complex disassembly. The work by Winter et al. shows that a hydrophobic loop of αSNAP is important for efficient disassembly, likely by membrane binding of SNAPs (Winter et al., 2009). We have indicated the location of the membrane in Figure 1C and clarified our point in the text.

11) The authors showed evidence that two αSNAP molecules bind to the interface between syntaxin and VAMP2, which is very interesting. The evidence is consistent with recent observations that αSNAP stabilizes zippered SNARE complex (Ma et al., 2016) and Sec17 promotes membrane fusion (Zick et al., 2015). However, the fact that NSF disassembles all SNARE complexes, mis-assembled SNARE complexes, and t-SNARE complexes (with a broken interface) indicates certain plasticity in binding between αSNAP and SNARE complexes, including binding stoichiometry and site (Vivona et al., 2013). Please discuss this feature.

We thank the reviewers for the comment. Indeed, different stoichiometries occur for different SNARE complexes, and this is discussed in the revised Discussion section and in the manuscript by White et al., (2018), which has been uploaded as a related manuscript:

“All structures of 20S complexes with full-length SNAP-25A determined to date, there are two, rather than four, bound αSNAPs. However, for SNARE complexes that do not contain a SNAP-25 linker such as the SNARE complex involved in vacuolar membrane fusion, the αSNAP number in the initial 20S complex may be higher (Lobingier et al., 2014). Moreover, additional αSNAP molecules may be engaged during the hydrolysis cycle (Shah et al., 2015).”

Additionally, charged surfaces of the SNAPs (positive) and SNARE complex (negative) likely underlie binding. Unlike a lock-and-key style model for binding specificity, an electrostatic model could accommodate a wide variety of SNARE complexes in various combinations and orientations. This model is supported by the charge mutations in αSNAP presented here as well as by detailed structural analysis of the electrostatic surfaces as presented in the White et al., (2018) manuscript.